# Distortion of AI Alignment: Does Preference Optimization Optimize for Preferences?

**Paul Gölz**
Cornell University
paulgoelz@cornell.edu

**Nika Haghtalab**
UC Berkeley
nika@berkeley.edu

**Kunhe Yang**
UC Berkeley
kunheyang@berkeley.edu

## Abstract

After pre-training, large language models are aligned with human preferences based on pairwise comparisons. State-of-the-art alignment methods (such as PPO-based RLHF and DPO) are built on the assumption of aligning with a single preference model, despite being deployed in settings where users have diverse preferences. As a result, it is not even clear that these alignment methods produce models that satisfy users *on average* — a minimal requirement for pluralistic alignment. Drawing on social choice theory and modeling users' comparisons through individual Bradley-Terry (BT) models, we introduce an alignment method's *distortion*: the worst-case ratio between the optimal achievable average utility, and the average utility of the learned policy. The notion of distortion helps draw sharp distinctions between alignment methods: *Nash Learning from Human Feedback* achieves the minimax optimal distortion of $(\frac{1}{2}+o(1)) \cdot \beta$ (for the BT temperature $\beta$), robustly across utility distributions, distributions of comparison pairs, and permissible KL divergences from the reference policy. RLHF and DPO, by contrast, suffer $\geq (1-o(1)) \cdot \beta$ distortion already without a KL constraint, and $e^{\Omega(\beta)}$ or even unbounded distortion in the full setting, depending on how comparison pairs are sampled.

## 1 Introduction

Reinforcement Learning from Human Feedback (RLHF) [38, 43, 18, 59] has become the dominant paradigm for aligning large language models (LLMs) with human values and preferences. In a typical alignment pipeline, human feedback is provided as ordinal comparisons between pairs of candidate model outputs. This feedback is used to fine-tune a pre-trained model, steering it toward the preferences expressed in these comparisons. A major limitation of RLHF and of many proposed alternatives (including DPO [43], ΦPO [5], KTO [24], SimPO [34], χPO [30]) is that they do not take into account that users will disagree on which model outputs are most useful or least harmful. A growing body of evidence — from both the general public and the research community [1, 47, 20, 8] — suggests that this blind spot of current alignment methods can lead to unfair outcomes. For example, Chakraborty et al. [13] argue that RLHF may align with a majority group's preferences and ignore the preferences of a minority.

In this work, we study a more basic question: *do current alignment methods reliably lead to a high average utility across the users?* Even such a minimal requirement might not be automatically met since alignment methods such as RLHF were originally designed with a single, perhaps representative, user in mind whose noisy ordinal preferences are assumed to be consistent with an underlying utility model. As a result, RLHF fits a single reward model to the observed ordinal comparisons of a population of users with different utility functions, effectively constructing a utility function for a "mythical" representative user. Could it be that optimizing a model for this mythical user leads to poor outcomes on average for real users? More fundamentally, *do ordinal preferences even contain enough information to ensure high average utility across a heterogeneous user population?*

39th Conference on Neural Information Processing Systems (NeurIPS 2025).

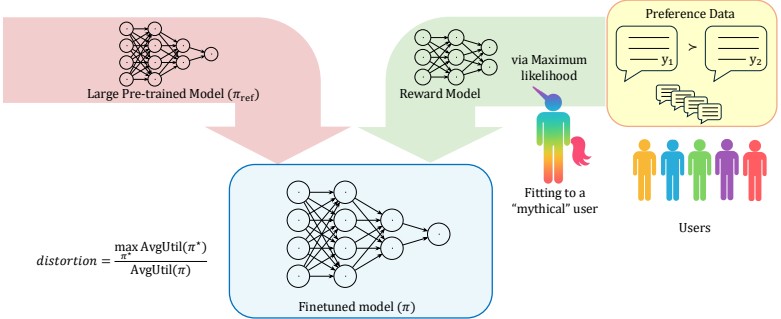

Figure 1: **The typical RLHF pipeline.** The preference optimization process begins by collecting comparison data from users with potentially diverse utilities. A single Bradley-Terry model is then fit to this data via Maximum Likelihood Estimation (MLE), producing a single reward model that represents a "mythical user" whose utility best explain the observed heterogeneous preferences. This reward model is used to fine-tune the pretrained policy. We define *distortion* as the ratio between the average utility of an optimal policy and that of the output policy, capturing how well the mythical user's utility aligns with the true average utility.

**Distortion of alignment.** To address these questions, we introduce the *distortion* of an alignment method,[1] which we define as the ratio between the optimal average utility of a policy (if the training process had access to users' true utilities) and the average utility achieved by the fine-tuned policy. A larger distortion implies lower average quality relative to the optimal policy. This notion is adapted from social choice theory [e.g., 41, 10, 4], where distortion quantifies the loss in average utility caused by using a voting rule that relies solely on ordinal preferences rather than full cardinal utilities.

Our setting departs from this classical formulation of distortion in two ways. First, we assume that users make pairwise comparisons probabilistically, following a Bradley–Terry model based on the user's idiosyncratic utilities. This assumption of probabilistic comparisons enables much less pessimistic distortion bounds than in the classic, deterministic-choice setting while capturing heterogeneous preferences. Second, our model and distortion bounds reflect that, in alignment, the models' generation policy is constrained to stay close to the pre-trained reference policy. These departures generate insights for both the social choice and the alignment communities.

## 1.1 Our Results

Our results address both the social choice setting with individual Bradley–Terry comparisons (Section 3) and the alignment setting (Section 4), which additionally constrains the policy to remain close in Kullback-Leibler (KL) divergence to a reference policy. The social choice setting is a special case of the alignment setting, in which the proximity constraint is not binding. Besides RLHF, which coincides with the *Borda* voting rule in the social choice setting, we study the proposed alternative *Nash Learning from Human Feedback (NLHF)* [36], which coincides with the *Maximal Lotteries* [25] voting rule. *Direct Preference Optimization (DPO)* [43] is equivalent to RLHF in our analysis and hence has the same distortion (see Appendix F.3). We define these alignment methods and voting rules in Section 2. In Section 5, we discuss how our results extend to KL-regularized (rather than constrained) alignment methods and to generalized models of sampling comparison pairs.

In this overview of results, summarized in Table 1, we present our bounds for the case where the number of sampled pairwise comparisons goes to infinity. In later sections, we accompany these statements with polynomially fast, finite-sample convergence bounds.

Our results establish that some distortion is unavoidable: in the social choice setting (i.e., without KL constraints), if each user only provides a single comparison, we show through a non-identifiability argument[2] that, for each value $\beta > 0$ of the Bradley–Terry temperature, *every* alignment method (or, equivalently, any voting rule) will suffer a distortion of $(\frac{1}{2} + o(1))\beta$ on some instances. This lower bound reflects a fundamental information bottleneck: even under Bradley–Terry generative

---

[1]While prior work has called for the study of distortion in alignment [21], or used alignment as a motivation for studying the distortion of voting rules [29, 22], our work is to our best knowledge the first to systematically define and analyze the distortion of alignment methods. We discuss these related efforts in Appendix A.

[2]While the non-identifiability of mixtures of ranking models is well established [57, 56, 55], our result shows that the non-identifiability can be *catastrophic*, leading to unavoidable loss in average utility.

Table 1: Overview of distortion bounds by alignment method and setting.

| Alignment Method | Social Choice Setting | AI Alignment Setting |
|---|---|---|
| RLHF [59] | $\leq O(\beta^2)$ [Thm 2] $\geq (1-o(1))\beta$ [Thm 5] *(Borda)* | $\geq e^{\Omega(\beta)}$ [Thm 6] / Unbounded in $\beta^*$ [Thm 9] |
| NLHF [36] | $= \left(\frac{1}{2}+o(1)\right)\beta$ [Cor 4] *(Max. Lotteries)* | $= \left(\frac{1}{2}+o(1)\right)\beta$ [Thm 7] |
| all *(one comparison per user/ Condorcet loser property)* | $\geq \left(\frac{1}{2}+o(1)\right)\beta$ [Thm 3] | $\geq \left(\frac{1}{2}+o(1)\right)\beta$ [Thm 3] |

[*]comparison pairs sampled from a distribution over pairs.

assumptions, ordinal feedback is not rich enough to perfectly optimize for the average utility of a heterogeneous user population. If each user provides $d \geq 2$ (possibly correlated) comparisons, the same lower bound applies to all voting rules that satisfy a probabilistic relaxation of the *Condorcet loser criterion*, a social-choice axiom widely satisfied by desirable voting rules, including Borda and Maximal Lotteries. As a result, this lower bound extends to RLHF and NLHF.

In the social choice setting, we show that both Borda and Maximal Lotteries have a distortion that is bounded in $\beta$. Borda's distortion lies between $(1-o(1))\beta$ and $O(\beta^2)$, whereas Maximal Lotteries' distortion is $(\frac{1}{2}+o(1))\beta$, matching even the lower-order terms of the lower bound. These results are of independent interest to the social choice community since they show that the introduction of randomized pairwise comparisons circumvents the necessary growth of distortion in the number of alternatives $m$. Recently, Goyal and Sarmasarkar [29] showed that the same Bradley–Terry assumption can reduce the distortion of specific voting rules in the metric distortion setting, where utilities are distances in a metric space. Our results show that the Bradley–Terry assumption has an even larger impact on general-utility distortion, where constant distortion is classically impossible, than in metric distortion. We derive the distortion bounds through a simple yet broadly applicable linearization lemma that sandwiches win-rates between linear functions, which can be generalized to other random utility models as well. These distortion bounds also carry implications for AI leaderboards such as Chatbot Arena [16], where heterogeneous user preferences across diverse tasks are aggregated via MLE under a single Bradley–Terry model — effectively equivalent to using Borda scores. We elaborate on these implications in Section 3.3.

In the alignment setting, we show that NLHF maintains Maximal Lotteries' optimal $(\frac{1}{2}+o(1))\beta$ distortion with remarkable robustness: regardless of the population's utilities, how comparison pairs are sampled, the number of comparisons per user, the reference policy, and the bound on the permissible KL divergence, NLHF obtains a $\Omega(1/\beta)$ fraction of the highest average utility achievable within the KL-divergence bound. Though we present this result for KL-constrained NLHF, we show in Section 5 that this directly implies a similar distortion guarantee for *regularized* NLHF. In contrast, RLHF's distortion can grow as $e^{\Omega(\beta)}$ in the alignment setting and is even unbounded in $\beta$ if the two outcomes to be compared are sampled in a correlated way rather than i.i.d.

We discuss additional related work in Appendix A.

## 2 Preliminaries

Let $A = \{1, \ldots, m\}$ be a finite set of *alternatives*. The population of users is described by a probability distribution $\mathcal{D}$ over *utility vectors* $u = (u(1), \ldots, u(m))$, whose entries $0 \leq u(x) \leq 1$ indicate a user's utility for alternative $x$.[3] The objective is to find an alternative $x$ such that its *average utility* $\mathsf{AvgUtil}(x) := \mathbb{E}_{u \sim \mathcal{D}}[u(x)]$ across the user population (also known as the *utilitarian social welfare*) is as high as possible. We extend this notation to probability distributions $\pi$ over alternatives by setting $\mathsf{AvgUtil}(\pi) := \mathbb{E}_{x \sim \pi}[\mathsf{AvgUtil}(x)]$.

Both voting rules and alignment methods observe comparisons from $n$ users. We model each $i = 1, \ldots, n$ as a fresh user with independently drawn utility vector $u_i \sim \mathcal{D}$. For exposition, we assume that each user $i$ provides an equal number $d \geq 1$ of pairwise comparisons. For each $i$, and for $j = 1, \ldots, d$, we independently draw alternatives $x_i^j, y_i^j$ from a fixed distribution $\mu$ over

---

[3]Our assumption that utilities be in $[0, 1]$ is weaker than any of the three assumptions — unit-sum, unit-range, or approval [23] — made in classic distortion to avoid trivial infinite lower bounds.

alternatives, in which the minimum probability mass $\mu_{\min} := \min_{x \in A} \mu(x)$ is positive.[4] User $i$ then compares each pair $\{x_i^j, y_i^j\}$ (for $j = 1, \ldots, d$) through a Bradley-Terry model based on $i$'s utilities: $i$ prefers $x_i^j$ over $y_i^j$ (written "$x_i^j \succ_i y_i^j$") with probability $\sigma\big(\beta \cdot (u_i(x_i^j) - u_i(y_i^j))\big)$, where $\sigma(t) := 1/(1+e^{-t})$ is the logistic sigmoid function and $\beta > 0$ is a temperature parameter, and prefers $y_i^j$ over $x_i^j$ ("$y_i^j \succ_i x_i^j$") otherwise.[5] Whereas this specifies the marginal probability of each pairwise comparison, we make no assumption about the correlation between $i$'s choices. For example, $i$ might derive the pairwise comparisons from a Plackett-Luce ranking, ensuring that the user's comparisons are always consistent.[6] We set $p(x \succ y)$ for the expected win rate $\mathbb{E}_{u \sim \mathcal{D}}\big[\sigma\big(\beta \cdot (u(x) - u(y))\big)\big]$.

**Social Choice Setting.** A *voting rule* $f$ observes the sampled pairwise comparisons $\{x_i^j \succ_i y_i^j\}_{i \in [n], j \in [d]}$ and maps them to a probability distribution over alternatives. For some $m, \mathcal{D}, \beta, d, \mu$, and the correlation between comparisons, the *average utility* of $f$ for $n$ samples is $\mathsf{AvgUtil}_n(f) := \mathbb{E}\big[\mathsf{AvgUtil}\big(f(\{x_i^j \succ_i y_i^j\}_{i,j})\big)\big]$, where the expectation is taken over the pairwise comparisons. The *distortion* of $f$ on $\mathcal{D}$ is the competitive ratio between $\mathsf{AvgUtil}_n(f)$ and the optimal average utility $\max_{x \in A} \mathsf{AvgUtil}(x)$ in the limit of $n \to \infty$ samples, and the distortion of $f$ the worst-case distortion over all $\mathcal{D}$:

$$dist(f, \mathcal{D}) := \limsup_{n \to \infty} \frac{\max_{x \in A} \mathsf{AvgUtil}(x)}{\mathsf{AvgUtil}_n(f)}, \qquad dist(f) = \sup_{\mathcal{D}} dist(f, \mathcal{D}).$$

For alternatives $x, y$, let $\#(x \succ y) := \{(i, j) \mid x_i^j = x, y_i^j = y\}$ denote the number of pairwise comparisons in which $x$ beat $y$. The *(normalized) Borda score* [45] of alternative $x$ is

$$\mathsf{BC}(x) := \frac{\sum_{y \in A} \#(x \succ y)}{\sum_{y \in A} \#(x \succ y) + \sum_{y \neq x} \#(y \succ x)},$$

i.e., the fraction of pairwise comparisons involving $x$ in which it wins. The *Borda* voting rule chooses the winner uniformly among all alternatives with maximum Borda score. The *Maximal Lotteries* voting rule first computes the margin matrix $M \in \mathbb{R}^{m \times m}$, where $M_{x,y} = \frac{\#(x \succ y) - \#(y \succ x)}{\#(x \succ y) + \#(y \succ x)}$. It then considers a symmetric two-player zero-sum game in which player 1 selects alternative $x_1$, player 2 selects alternative $x_2$, and the payoffs are $M_{x_1,x_2}$ for player 1 and $M_{x_2,x_1} = -M_{x_1,x_2}$ for player 2. The maximal lotteries rule returns a distribution $\pi \in \mathrm{argmax}_{\pi_1 \in \Delta(A)} \min_{\pi_2 \in \Delta(A)} \mathbb{E}_{x_1 \sim \pi_1, x_2 \sim \pi_2}[M_{x_1,x_2}]$, i.e., a mixed strategy in Nash equilibrium. When several such $\pi$ exist, our results hold for any choice.

**Alignment Setting.** The alignment setting generalizes the social choice setting in two ways: first, user utilities $u_i(y \mid \boldsymbol{x})$ may depend on a state $\boldsymbol{x}$; second, the goal in determining a policy $\pi$ is not purely to maximize the reward $\mathsf{AvgUtil}(\pi \mid \boldsymbol{x})$, but a trade-off between this reward and the goal of remaining close to a reference policy $\pi_{\mathrm{ref}}(\cdot \mid \boldsymbol{x}) \in \Delta(A)$ in terms of the KL divergence $D_{\mathrm{KL}}(\cdot \| \pi_{\mathrm{ref}})$. For theoretical tractability, we focus our analysis on a single state $\boldsymbol{x}$, which we from here on omit from the notation. Conceptually, this treatment of alignment on a state-by-state basis corresponds to an assumption that our policy class is expressive enough so that it can take the optimal distribution of actions at each state[7] and abstracts from the generalization problem of estimating the population's preference between a pair of alternatives at the given state $\boldsymbol{x}$ based on preferences in similar states $\boldsymbol{x}'$.

Having set aside the dependency between states, we focus on how the regularization with respect to a reference policy impacts the ability to optimize the average utility of three alignment methods: RLHF, DPO, and NLHF. In addition to the pairwise comparisons, these methods take in a *reference policy* $\pi_{\mathrm{ref}} \in \Delta(A)$ and a *KL bound* $\tau \geq 0$, and map these inputs to a policy $\pi$ in the *KL-ball* $B_\tau(\pi_{\mathrm{ref}}) := \{\pi \in \Delta(A) \mid D_{\mathrm{KL}}(\pi \| \pi_{\mathrm{ref}}) \leq \tau\}$ around the reference policy. RLHF, DPO, and NLHF are typically implemented with a KL regularization rather than our KL-constrained formulation, which we adopt to enable a comparison on equal terms. In Section 5, we show that these perspectives are equivalent, and that distortion upper bounds carry over to regularized alignment methods.

---

[4]In Section 5, we discuss how most of our results extend more general distributions over comparison pairs, which can, for example. capture $k$-wise comparisons.

[5]Should we sample the same alternative $x = x_i^j = y_i^j$ twice for a pair, the user is not asked for a pairwise comparison. We record this as "$x \succ_i x$", in a slight abuse of notation.

[6]In particular, if we sample the same unordered pair twice for a user, the answers can be perfectly correlated.

[7]This assumption is common in the literature; see, for example, Rafailov et al. [43]'s application of first-order optimality conditions of PPO loss minimization.

The *RLHF method* first estimates rewards for each alternative, using maximum likelihood estimation assuming that comparisons were generated by a single Bradley–Terry model:

$$r := \text{argmax}_{r \in \mathbb{R}^m} \sum_{1 \le i \le n, 1 \le j \le d} \log\left(\sigma(r(x_i^j) - r(y_i^j))\right).^8$$

Next, RLHF uses PPO [44] to compute the policy $\pi_{\text{RLHF}} := \text{argmax}_{\pi \in B_\tau(\pi_{\text{ref}})} \mathbb{E}_{x \sim \pi}[r(x)]$ with maximum expected reward within the KL-ball. In our setup, DPO is equivalent to RLHF (see Appendix F.3) and thus has the same distortion.

The alignment method NLHF was inspired in part by a desire to better align with the preferences of a heterogeneous group [36]. NLHF naturally adapts the definition of maximal lotteries by constraining both players' mixed strategies to the KL-ball, i.e., $\pi_{\text{NLHF}} := \text{argmax}_{\pi_1 \in B_\tau(\pi_{\text{ref}})} \min_{\pi_2 \in B_\tau(\pi_{\text{ref}})} \mathbb{E}_{x_1 \sim \pi_1, x_2 \sim \pi_2}[M_{x_1, x_2}]$.

To generalize the definition of distortion to alignment methods, we set the maximum average utility of any policy in the KL-ball as the benchmark. For fixed $m, \mathcal{D}, \beta, d, \mu$ and correlation between pairwise comparisons, the distortion of alignment method $f$ is

$$dist(f) = \sup_{\mathcal{D}, \pi_{\text{ref}}, \tau} \limsup_{n \to \infty} \frac{\max_{\pi \in B_\tau(\pi_{\text{ref}})} \text{AvgUtil}(\pi)}{\text{AvgUtil}_n(f(\cdot, \pi_{\text{ref}}, \tau))}.$$

## 3 Social Choice (or AI Alignment without KL Constraint)

We begin the demonstration of our distortion framework in the social choice setting. From the perspective of alignment, this setting is the limit where the KL constraint (equivalently, KL regularization) to the reference policy vanishes. Hence, distortion measures whether the "direction" in which an alignment method pushes the pre-trained policy is aligned with average utility at all.

Moreover, the social choice setting allows us to illustrate how the Bradley-Terry assumption overcomes the pessimism of classic deterministic-choice distortion. In the classic setting, high distortion — $\Omega(\sqrt{m})$ even for randomized voting rules and under utility-normalization assumptions [10, 23] — is unavoidable because a voting rule observes no signal about *preference intensity*, i.e., whether a user prefers $a$ over $b$ strongly or is merely breaking a tie between equally valued alternatives. Random Bradley-Terry comparisons would clearly side-step this problem if we could observe many samples of each pairwise comparison *for a single utility vector*: by consistency, the Bradley-Terry MLE would recover the utilities (up to an additive shift), allowing us to select the utility-maximizing alternative and achieve a perfect distortion of 1.

It is not obvious, by contrast, that random pairwise comparisons will be similarly useful in our heterogeneous setting, where each observation is drawn from a mixture of users' Bradley-Terry models. Because users are not labeled and may provide as little as a single pairwise comparison, there is no hope to cluster users and estimate rewards per cluster. Instead, a source of inspiration is an observation by Caragiannis and Procaccia [12] in a much simpler model, in which each user votes for a single alternative with probability equal to their utility (which is normalized to sum to 1). Since the probability of the event "$i$ votes for $x$" equals $u_i(x)$, the total number of votes of alternative $x$ (i.e., $\sum_{i \in N} \mathbb{1}_{i \text{ votes for } x}$) is an unbiased estimator of its total utility. For many samples, this estimator concentrates around its mean and allows to select the optimal alternative.

The argument would extend if some observed events from a user had a probability that is affine in the user's utilities. Alas, we are not so lucky: the sigmoid function in the probability of the event "$i$ ranks $x$ over $y$" is nonlinear, and we show in Theorem 3 that this nonlinearity makes a distortion of at least $\frac{\beta}{2} \frac{1 + e^{-\beta}}{1 - e^{-\beta}} > 1$ unavoidable.

Though our observations' probabilities are not affine in the utilities, we can bound these probabilities by affine functions, which ultimately powers our distortion upper bounds. As shown in Fig. 2, we sandwich the probability $\sigma(\beta \cdot (u(x) - u(y)))$ that a user with utilities $u$ prefers $x$ over $y$ between the affine lower bound $\beta \cdot (L u(x) - \ell_\beta u(y)) + \frac{1}{2}$ and affine

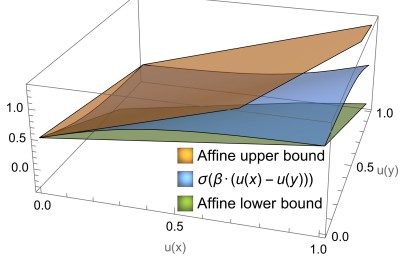

Figure 2: Bounds on probability of preferring $x$ over $y$, $\beta = 5$.

$^8$Assuming that each $x \in A$ wins at least one pairwise comparison against each $y \ne x$, this optimization has a maximizer, which is unique up to additive shifts, by strict convexity.

upper bound $\beta \cdot (\ell_\beta\, u(x) - L\, u(y)) + \frac{1}{2}$, for constants $L, \ell_\beta$ defined below. By linearity, this bound extends to a bound of the expected win-rate $p(x \succ y) = \mathbb{E}_{u \sim \mathcal{D}}\left[\sigma(\beta \cdot (u(x) - u(y)))\right]$ by affine expressions in $\mathsf{AvgUtil}(x) = \mathbb{E}_{u \sim \mathcal{D}}\left[u(x)\right]$ and $\mathsf{AvgUtil}(y) = \mathbb{E}_{u \sim \mathcal{D}}\left[u(y)\right]$. We defer the lemma's formal proof to Appendix C.

**Lemma 1** (Linearization of Expected Win-Rates). *Let $L := \sigma'(0) = 1/4$ and $\ell_\beta := \frac{\sigma(\beta) - \frac{1}{2}}{\beta} = \frac{1}{2\beta} \cdot \frac{1 - e^{-\beta}}{1 + e^{-\beta}}$. For any pair of alternatives $x, y \in A$, we have*

$$\beta \cdot (\ell_\beta \cdot \mathsf{AvgUtil}(x) - L \cdot \mathsf{AvgUtil}(y)) \leq p(x \succ y) - \tfrac{1}{2} \leq \beta \cdot (L \cdot \mathsf{AvgUtil}(x) - \ell_\beta \cdot \mathsf{AvgUtil}(y)).$$

### 3.1 Upper Bound on Borda Distortion

Siththaranjan et al. [46] observed that RLHF and the Borda voting rule are closely linked in that the Bradley-Terry MLE rewards are ordered by their alternatives' Borda score. Hence, as the KL constraint relaxes, RLHF moves all of the policy's probability mass on the Borda winner. Because Borda has infinite distortion in the classic setting [41], we would hope that distortion is more reasonable under our assumptions. Fortunately, Borda indeed has at most $O(\beta^2)$ distortion, which we prove through two applications of our linearization lemma:

**Theorem 2** (Borda Distortion Upper Bound). *For any instance $\mathcal{D}$ with any number of alternatives $m$, distribution $\mu$ over alternatives, and temperature $\beta$, Borda has at most distortion $\left(\frac{\beta}{2} \cdot \frac{1 + e^{-\beta}}{1 - e^{-\beta}}\right)^2 = O(\beta^2)$. In the finite-sample regime, we have that*

$$\mathsf{AvgUtil}_n(\mathsf{Borda}) \geq \left(\tfrac{2}{\beta} \cdot \tfrac{1 - e^{-\beta}}{1 + e^{-\beta}}\right)^2 \cdot \max_{x^\star \in A} \mathsf{AvgUtil}(x^\star) - O\left(\tfrac{1}{\beta^2}\sqrt{\tfrac{\log(mn\beta)}{n \cdot \min\{1, d\mu_{\min}^2\}}} + \tfrac{m \log(mn\beta)}{n \cdot \beta^2 \mu_{\min}^2}\right).$$

*Proof sketch (full proof in Appendix E.1).* For exposition, we sketch this proof for $n \to \infty$, assuming that each alternative's Borda count has converged to its expectation $\mathsf{BC}^\star(x) := \sum_{y \in A} \mu(y) \cdot p(x \succ y)$. Let $\hat{x} = \arg\max_{x \in A} \mathsf{BC}^\star(x)$ be the Borda winner in the limit, and $x^\star = \arg\max_{x \in A} \mathsf{AvgUtil}(x)$ be the utility maximizer. Applying Lemma 1 to the win-rates $p(\hat{x} \succ y)$ and $p(x^\star \succ y)$ for all $y \in A$, we have that

$$\begin{aligned} \mathsf{BC}^\star(\hat{x}) - \tfrac{1}{2} &\leq \sum_{y \in A} \mu(y) \cdot \beta \left(L \cdot \mathsf{AvgUtil}(\hat{x}) - \ell_\beta \cdot \mathsf{AvgUtil}(y)\right) \\ &= \beta \left(L \cdot \mathsf{AvgUtil}(\hat{x}) - \ell_\beta \cdot \mathsf{AvgUtil}(\mu)\right); \\ \mathsf{BC}^\star(x^\star) - \tfrac{1}{2} &\geq \sum_{y \in A} \mu(y) \cdot \beta \left(\ell_\beta \cdot \mathsf{AvgUtil}(x^\star) - L \cdot \mathsf{AvgUtil}(y)\right) \\ &= \beta \left(\ell_\beta \cdot \mathsf{AvgUtil}(x^\star) - L \cdot \mathsf{AvgUtil}(\mu)\right). \end{aligned} \tag{1}$$

Since $\mathsf{BC}^\star(\hat{x}) \geq \mathsf{BC}^\star(x^\star)$, we obtain from the above two inequalities that

$$L \cdot \mathsf{AvgUtil}(\hat{x}) + (L - \ell_\beta) \cdot \mathsf{AvgUtil}(\mu) \geq \ell_\beta \cdot \mathsf{AvgUtil}(x^\star). \tag{2}$$

A standard averaging argument shows that $\mathbb{E}_{x \sim \mu}[\mathsf{BC}^\star(x)] \geq 1/2$, which implies that $\mathsf{BC}^\star(\hat{x})$ must be at least $1/2$. Combining this with Eq. (1), we obtain that $\mathsf{AvgUtil}(\mu) \leq \frac{L}{\ell_\beta} \cdot \mathsf{AvgUtil}(\hat{x})$. Substituting this into Equation (2) (noting that $L \geq \ell_\beta$), we have that

$$\begin{aligned} \tfrac{L^2}{\ell_\beta}\mathsf{AvgUtil}(\hat{x}) &= L \cdot \mathsf{AvgUtil}(\hat{x}) + \tfrac{(L - \ell_\beta)L}{\ell_\beta} \cdot \mathsf{AvgUtil}(\hat{x}) \\ &\geq L \cdot \mathsf{AvgUtil}(\hat{x}) + (L - \ell_\beta) \cdot \mathsf{AvgUtil}(\mu) \geq \ell_\beta \cdot \mathsf{AvgUtil}(x^\star), \end{aligned}$$

which yields the distortion guarantee $\frac{\mathsf{AvgUtil}(x^\star)}{\mathsf{AvgUtil}(\hat{x})} \leq \left(\frac{L}{\ell_\beta}\right)^2 = \left(\frac{\beta}{2}\frac{1 + e^{-\beta}}{1 - e^{-\beta}}\right)^2$. $\qquad\square$

### 3.2 Lower Bounds (and Upper Bound for Maximal Lotteries)

The upper bound for Borda nested two applications of the linearization lemma. As a result, it twice incurred a distortion factor of $\frac{L}{\ell_\beta} = \frac{\sigma'(0)}{(\sigma(\beta) - \sigma(0))/\beta}$, which measures the sigmoid function's deviation from linearity in the relevant range. Below, we show that *any* voting rule must incur this factor at least once, at least for the case of $d = 1$ comparisons per user. This distortion occurs even though the voting rule has access to infinitely many pairwise comparison samples, which shows that the nonlinearity of the Bradley-Terry model can cause a loss of the information necessary to find the utility maximizer.

The proof (in Appendix E.3) constructs a user population $\mathcal{D}$ in which a small minority has utility 1 for some special alternative $a$ and 0 for all other alternatives, whereas the majority has a small utility $\epsilon$ for all alternatives except for $a$, for which they have utility 0. The sizes of these blocs are balanced such that all expected win-rates are $1/2$. Due to the diminishing returns in the sigmoid function, the resulting average utility for $a$ is $\frac{L}{\ell_\beta}$ times higher than that of the other alternatives. But since the pairwise comparisons observed by a voting rule are just independent Bernoulli draws with bias $1/2$, all versions of the instance with permuted alternatives are indistinguishable. Since no voting rule can identify alternative $a$ better than random guessing, they must incur $\frac{L}{\ell_\beta}$ distortion.

If there are $d \geq 2$ observations per user, the above argument does not apply to all voting rules because an elaborate voting rule might use the correlations within a user's comparisons to identify $a$. We can, however, extend the lower bound to $d \geq 2$ for all voting rules that put at most $1/m$ probability mass on a *Condorcet loser*, i.e., an alternative $x$ such that $\#(y \succ x) > \#(x \succ y)$ for all $y \neq x$. This property generalizes the *Condorcet loser criterion* and is satisfied by a wide range of voting rules deemed desirable, including Borda and Maximal Lotteries. The proof uses essentially the same instance as above, slightly tipping the expected win-rates against $a$ to make it a Condorcet loser. Since the social choice setting is a special case of alignment, the lower bound extends to alignment.

**Theorem 3** (Voting Rule-Independent Distortion Lower Bound). *Fix any $\beta > 0$. If each user provides $d{=}1$ comparison, no voting rule can guarantee distortion better than $\frac{\beta}{2} \cdot \frac{1+e^{-\beta}}{1-e^{-\beta}}$ for large $m$. If each user reports $d \geq 2$ pairwise comparisons, any voting rule that puts at most $1/m$ probability mass on a Condorcet loser must have at least the above distortion.*

The Maximal Lotteries voting rule exactly matches the above lower bound of $\frac{\beta}{2} \cdot \frac{1+e^{-\beta}}{1-e^{-\beta}}$. We omit the proof here, as it follows as a direct corollary of NLHF's upper bound (Theorem 7) in the next section.

**Corollary 4** (of Theorem 7). *The Maximal Lotteries voting rule has a distortion of $\frac{\beta}{2} \cdot \frac{1+e^{-\beta}}{1-e^{-\beta}}$.*

The Borda rule, in contrast, does not match this optimal distortion bound, as shown by the following bound that we prove and state in full detail in Appendix E.2.

**Theorem 5** (Borda Distortion Lower Bound, Informally). *For any $\beta > 0$ and $m \geq 3$, the distortion guaranteed by Borda (and, hence, RLHF) is greater than and bounded away from $\frac{\beta}{2} \frac{1+e^{-\beta}}{1-e^{-\beta}}$. In particular, as $\beta \to \infty$, this distortion guarantee is at least $(1 - o(1)) \cdot \beta$.*

### 3.3 Discussion

**Implications for Social Choice Theory.** Though we have presented these bounds in terms of their implications for alignment, they are of independent interest to social choice theory. In the *classical* social choice setting — where each voter's ranking is produced deterministically, without any randomness such as that introduced by a Bradley–Terry model — the distortion framework (with nonnegative utilities) exhibits serious limitations. In particular, it leads to unreasonably high distortion and some unnatural prescriptions. For example, any deterministic voting rule has distortion $\Omega(m^2)$, where optimal distortion is achieved by Plurality [12] (a rule widely disregarded by social choice theorists) whereas Borda and all Condorcet consistent rules have infinite distortion [41]. These limitations may explain in part why recent research activity [e.g., 28, 14, 29] has focused on the *metric distortion* setting [3, 4], in which many natural voting rules have constant distortion. But this comes at the cost of expressiveness: the metric setting assumes utilities are (negated) distances satisfying the triangle inequality. For example, the metric distortion setting implies that, if $i$ has high utility for $x$ and $y$, and $j$ has high utility for $x$, then $j$ must also have high utility for $y$, which need not be the case in our setting. We see our assumption of a user-specific random choice model as another way to make distortion a more practical criterion for choosing between voting rules.

**Implications for AI Leaderboards.** Our results also have implications beyond being a special case of alignment. A notable example is LLM leaderboards such as Chatbot Arena [16], where users submit prompts, are shown the responses of two anonymized models, and select their preferred response. The leaderboard aggregates these pairwise comparisons by fitting a Bradley-Terry model via MLE, and ranking the models according to their estimated rewards. As in the alignment setting, this approach assumes a single latent notion of LLM quality, ignoring the fact that LLMs are used by diverse users for a wide range of tasks, each with their own goals, preferences, and prompt styles. This setting fits neatly into our social-choice model, where $\mathcal{D}$ captures a random user's utility for the responses of different models to a random prompt (drawn from an arbitrary joint distribution over users and prompts), and AvgUtil quantified the average utility a model delivers for a random user and task, which captures the model's *usability*. We defer a more detailed discussion to Appendix B.

# 4   AI Alignment with KL Constraint

We now tackle the general alignment setting, in which the output policy $\pi$ must be chosen within a prescribed KL divergence of the reference policy $\pi_{\text{ref}}$. This setting is more challenging than the social choice setting because even an alignment method that would choose high-utility alternatives in the absence of constraints might make poor use of a finite KL budget.

## 4.1   Lower Bound for RLHF

Before presenting the optimal distortion upper bound for NLHF, we illustrate the pitfalls of the alignment setting with a lower bound on RLHF. This bound shows that a KL constraint can cause RLHF to have exponential distortion in $\beta$, exceeding its quadratic upper bound in the social choice setting (Theorem 2).

**Theorem 6** (RLHF Distortion Lower Bound). *For $m \geq 3$, there is a sequence of alignment problems on which the distortion of RLHF scales as $e^{\Omega(\beta)}$ in $\beta$.*

*Proof sketch (full proof in Appendix F.2).* For ease of exposition, consider an instance with three alternatives $a, b, c$ where $\mu(c)$ is about $e^{\beta}$ times larger than $\mu(a) = \mu(b)$.[9] Let the population consist of a tiny minority (a $\Theta(e^{\beta})$ fraction) with utilities $u(a) = 0, u(b) = 1, u(c) = 0$, and a large majority with utilities $u(a) = \frac{1}{\beta}, u(b) = 0, u(c) = 1$. Both $a$ and $b$ are likely to be beaten by $c$, but by carefully choosing the size of the minority, we can make $p(b \succ c) > p(a \succ c)$, i.e., we can make $b$'s advantage of being preferred by the minority outweigh $a$'s advantage of being slightly less dispreferred by the majority. Since $\mu(c)$ is so much larger than $\mu(a), \mu(b)$, the vast majority of pairwise comparisons involving $a$ or $b$ are against $c$. As a result, the MLE reward for $b$ will be higher than for $a$, even though $\mathsf{AvgUtil}(a) = \Theta(\frac{1}{\beta})$ is exponentially larger than $\mathsf{AvgUtil}(b) = \Theta(e^{-\beta})$. (In the social choice setting, this would not be a problem because $c$ has even higher average utility and higher reward.)

The lower bound arises for a reference policy that puts a tiny probability mass $\varepsilon$ on $c$, and $\frac{1-\varepsilon}{2}$ probability mass each on $a$ and $b$, together with a KL constraint of $\tau = \log 2$. Now $D_{\text{KL}}(\pi \,\|\, \pi_{\text{ref}}) = \pi(a) \log \frac{\pi(a)}{(1-\varepsilon)/2} + \pi(b) \log \frac{\pi(b)}{(1-\varepsilon)/2} + \pi(c) \log \frac{\pi(c)}{\varepsilon}$. Since $\varepsilon$ is very small, $\pi(c)$ cannot be increased by enough to make a meaningful difference on the achievable utility; but the KL budget essentially allows to spread the probability mass of $\pi$ freely between $a$ and $b$. Since $b$ has a higher MLE reward, RLHF puts almost all of $\pi$'s mass on $b$, which yields exponentially less utility than the utility-maximizing policy in the KL ball, which puts almost all mass on $a$.   $\square$

This bound formalizes a key limitation of the *reward-based* approach inherent to RLHF. The MLE phase of RLHF attempts to fit rewards to the observed comparisons, whose frequencies are determined by $\mu$. Due to preference heterogeneity, not all three pairwise win-rates can be simultaneously fit by a reward vector, so the MLE sacrifices accuracy on the rarely observed pair $\{a, b\}$ for higher accuracy of comparisons involving $c$. By placing so little mass on $c$, our choice of $\pi_{\text{ref}}$ forces RLHF to choose between the misrepresented alternatives $a$ and $b$, causing it to make a high-distortion choice.

Our lower bound exploits that the distribution $\mu$ governing the frequencies of comparison pairs differs greatly from the reference policy $\pi_{\text{ref}}$. We leave open to characterize RLHF's distortion under the assumption that $\mu = \pi_{\text{ref}}$, for which we only know the lower bound Theorem 5.

## 4.2   Distortion of NLHF

While we saw above that a mismatch between input distribution $\mu$ and reference policy $\pi_{\text{ref}}$ can lead RLHF towards highly suboptimal policies, NLHF has no such problem. Below, we show that, across all settings of our model, NLHF's distortion exactly matches the lower bound from Theorem 3.

Despite the generality of this result, the proof is no harder than our upper bound for Borda in the social choice setting and involves only a single application of the linearization lemma. It also highlights a key advantage over RLHF's reward-based approach: Since the NLHF policy is computed as a Nash-equilibrium strategy in a game where the opponent might select any policy in the KL ball, the NLHF automatically "hedges" to perform well in expectation against all such policies, including the utility-maximizing benchmark $\pi^{\star}$. Since the social choice setting is a special case of alignment, this theorem immediately implies the distortion upper bound for Maximal Lotteries (Corollary 4), and both NLHF and Maximal Lotteries are minimax optimal by the lower bound in Theorem 3.

---

[9]To avoid such unbalanced $\mu$, one could equivalently copy alternative $c$ many times and let $\mu$ be uniform.

**Theorem 7** (NLHF Distortion Upper Bound). *For any instance $\mathcal{D}$ and any $m$, data distribution $\mu$, temperature $\beta$ of the Bradley-Terry model, and any reference policy $\pi_{ref}$ and KL budget $\tau$, we have $dist(\mathsf{NLHF}) \leq \frac{\beta}{2} \cdot \frac{1+e^{-\beta}}{1-e^{-\beta}}$. In the finite-sample regime, we have*

$$\mathsf{AvgUtil}_n(\mathsf{NLHF}) \geq \left(\frac{2}{\beta} \cdot \frac{1-e^{-\beta}}{1+e^{-\beta}}\right) \cdot \max_{\pi^\star \in B_\tau(\pi_{ref})} \mathsf{AvgUtil}(\pi^\star) - O\left(\frac{1}{\beta}\sqrt{\frac{\log(mn)}{n \cdot \min\{1, d \cdot \mu_{\min}^2\}}} + \frac{\log(mn)}{n \cdot \beta \mu_{\min}^2}\right).$$

*Proof sketch (full proof in Appendix F.1).* For exposition, we assume that the NLHF method knows the expected win-rates $p(x \succ y)$, and defer the proof of finite-sample guarantees. Hence, the NLHF policy by definition satisfies

$$\pi_{\mathsf{NLHF}} \in \operatorname*{argmax}_{\pi_1 \in B_\tau(\pi_{ref})} \min_{\pi_2 \in B_\tau(\pi_{ref})} \mathbb{E}_{x_1 \sim \pi_1, x_2 \sim \pi_2}[p(x_1 \succ x_2) - p(x_2 \succ x_1)].$$

Since this describes a Nash-equilibrium strategy for a symmetric two-player zero-sum game, and any such game has value 0, it must hold that

$$\min_{\pi_2 \in B_\tau(\pi_{ref})} \mathbb{E}_{x_1 \sim \pi_{\mathsf{NLHF}}, x_2 \sim \pi_2}[p(x_1 \succ x_2) - p(x_2 \succ x_1)] = 0.$$

Plugging in $p(x_1 \succ x_2) - p(x_2 \succ x_1) = 2\, p(x_1 \succ y_1) - 1$, we obtain

$$\min_{\pi_2 \in B_\tau(\pi_{ref})} \mathbb{E}_{x_1 \sim \pi_{\mathsf{NLHF}}, x_2 \sim \pi_2}[p(x_1 \succ x_2) - 1/2] = 0.$$

Using the utility-maximizing policy $\pi^\star := \operatorname{argmax}_{\pi \in B_\tau(\pi_{ref})} \mathsf{AvgUtil}(\pi)$ for $\pi_2$, we obtain that $\mathbb{E}_{x_1 \sim \pi_{\mathsf{NLHF}}, x_2 \sim \pi^\star}[p(x_1 \succ x_2) - \frac{1}{2}] \geq 0$.

At this point, we upper bound the win-rate with the linearization lemma (Lemma 1), and obtain

$$0 \leq \mathbb{E}_{x_1 \sim \pi_{\mathsf{NLHF}}, x_2 \sim \pi^\star}\left[\beta \cdot (L \cdot \mathsf{AvgUtil}(x_1) - \ell_\beta \cdot \mathsf{AvgUtil}(x_2))\right]$$
$$= \beta \cdot (L \cdot \mathsf{AvgUtil}(\pi_{\mathsf{NLHF}}) - \ell_\beta \cdot \mathsf{AvgUtil}(\pi^\star)).$$

This implies that $\frac{\mathsf{AvgUtil}(\pi^\star)}{\mathsf{AvgUtil}(\pi_{\mathsf{NLHF}})} \leq \frac{L}{\ell_\beta} = \frac{\beta}{2} \cdot \frac{1+e^{-\beta}}{1-e^{-\beta}} = O(\beta)$, thus completing the proof. $\qquad\square$

The simplicity of the proof above also speaks to its generality. For instance, the only property of KL divergence we used was that the feasible region $B_\tau(\pi_{ref})$ is a closed convex set (to ensure the existence of a Nash equilibrium). Consequently, the distortion bound of Nash learning extends to other ways of constraining proximity to the reference policy, such as by $\chi^2$ divergence [30].

## 5   Extensions of the Model

**KL Constraints vs. Regularization.**    In our model, we defined alignment methods as taking in an explicit KL bound $\tau$ as an input parameter, which is convenient for comparing the policy against a fair benchmark. In practice, however, alignment methods such as RLHF, DPO, and NLHF are *regularized* rather than constrained in terms of their KL-divergence. For example, the PPO phase of RLHF finds a policy $\pi$ maximizing the regularized objective $\mathbb{E}_{x \sim \pi}[r(x)] - \lambda D_{\mathsf{KL}}(\pi \,\|\, \pi_{ref})$, and the payoff matrix in the game solved by NLHF is $\mathbb{E}_{x_1 \sim \pi_1, x_2 \sim \pi_2}[M_{x_1,x_2}] - \lambda D_{\mathsf{KL}}(\pi_1 \,\|\, \pi_{ref}) + \lambda D_{\mathsf{KL}}(\pi_2 \,\|\, \pi_{ref})$, where $\lambda \geq 0$ is a regularization parameter given to the alignment method instead of $\tau$.

We prove in Appendix F.4 that the regularized and constrained versions of RLHF and NLHF are equivalent. That is, each policy $\pi$ returned by the $\lambda$-regularized version of a method is optimal for the $\tau$-constrained version for $\tau = D_{\mathsf{KL}}(\pi \,\|\, \pi_{ref})$ (and any policy returned by the $\tau$-constrained version is optimal for the $\lambda$-regularized version and some $\lambda \geq 0$).

Through this equivalence, any distortion upper bound in our setting applies to the KL-regularized versions of the alignment method: if $\pi$ results from the $\lambda$-regularized alignment method, $\pi$ is optimal for the $\tau = D_{\mathsf{KL}}(\pi \,\|\, \pi_{ref})$-constrained version by equivalence, at which point the distortion upper bound shows that $\pi$ can compete with any policy with no larger KL divergence from $\pi_{ref}$.[10] Applying this observation to Theorem 7, we obtain the following guarantee for regularized NLHF:

**Corollary 8.** *If $\lambda$-regularized NLHF (for any $\lambda \geq 0$) returns a policy $\tilde{\pi}_{\mathsf{NLHF}}$, this policy's average utility is at least a $\frac{2}{\beta} \cdot \frac{1-e^{-\beta}}{1+e^{-\beta}}$ fraction of the optimal average utility of any policy $\pi$ with $D_{KL}(\pi \,\|\, \pi_{ref}) \leq D_{KL}(\tilde{\pi}_{\mathsf{NLHF}} \,\|\, \pi_{ref})$ (minus finite-sample errors, see Theorem 7).*

---

[10]Why not define distortion by flexibly selecting the benchmark based on the output policy's KL divergence? For this definition, an alignment method that always returns the reference policy would spuriously achieve distortion 1: because its KL divergence is 0, we would benchmark it only against the reference policy, i.e., itself.

**Sampling of Comparison Pairs.** In our model, we assume that each voter provides $d$ pairwise comparisons, where both members $x_i^j, y_i^j$ of each comparison pair are sampled i.i.d. from $\mu$. More generally, we can model $\{x_i^j, y_i^j\} \sim \nu$ where $\nu$ is a distribution over unordered alternative pairs, or even a distribution over $d$ pairs of alternatives from which $\{x_i^1, y_i^1, \dots, x_i^d, y_i^d\}$ are sampled. (To keep the alignment methods well defined, we assume that each comparison pair has positive probability of being sampled.) The latter of these models can, for example, express $k$-wise (rather than pairwise) comparisons, if $d = \binom{r}{2}$ are all pairs inside a randomly chosen set of $r$ alternatives. Almost all of our results continue to hold in these general models: the lower bound for all alignment methods that satisfy the Condorcet loser criterion in the social choice setting (Theorem 3), the exponential lower bound for RLHF (Theorem 6), and the upper bound for NLHF/Maximal Lotteries (Theorem 7)[11].

Given that our proofs continue to work out, the only "disadvantage" of these stronger models for sampling comparison pairs is that, without a distribution $\mu$, the Borda voting rule is no longer defined (and we see no obvious way to generalize the Borda–MLE equivalence [46, 42]). It seems that RLHF does not only become harder to analyze under these comparison-pair models, but actually performs worse: we show in Appendix G that RLHF can have a distortion that is not bounded in $\beta$ in these extended models, leading to an even clearer separation with NLHF.

**Theorem 9** (Unbounded Distortion of RLHF Under Correlated Sampling). *For any $\beta > 0$, there exists a sequence of alignment instances and distributions $\nu \in \Delta\left(\binom{A}{2}\right)$ over comparison pairs such that RLHF's distortion is unbounded.*

## 6 Discussion and Practical Takeaways

In this paper, we introduced the notion of distortion for AI alignment. We showed that one such alignment method, NLHF, obtains the optimal distortion guarantee of $\left(\frac{1}{2} + o(1)\right)\beta$. Putting this bound into perspective, if we assume that a user will rate a minimally preferred alternative over a maximally preferred alternative with $1\%$ probability, this suggests a value of $\beta = \log \frac{99\%}{1\%} \approx 4.60$ and a distortion guarantee of about $2.34$, which is a quite reasonable worst-case guarantee.

For the incumbent method, RLHF, our analysis gave more negative results. Its distortion was worse than NLHF's in the unconstrained setting, exponentially worse in the constrained setting, and unbounded if the comparison pairs are not drawn i.i.d.. Given the ubiquity of RLHF, it is a pressing open question to fully characterize its distortion (especially when $\mu$ coincides with $\pi_{\text{ref}}$) or find assumptions that guarantee a lower distortion. A major technical challenge is that bounding this distortion requires reasoning not only about the relative ordering of rewards but also their magnitudes.

Beyond these theoretical contributions, our results yields several practical insights:

**Limitations of reward models as evaluation metrics.** Our findings indicate that reward-model scores are unreliable proxies for human satisfaction, not only because of insufficient data diversity or scale, but also due to fundamental information-theoretic limits of cardinal-to-ordinal conversion. Due to this limitation, it may be valuable to complement reward-model evaluations with direct measurements of cardinal preferences, such as graded feedback or satisfaction scores.

**Sensitivity to post-training data distribution.** The performance of reward-based pipelines such as RLHF is sensitive to the sampling distribution of pairwise comparisons, while the reward-free method NLHF shows greater robustness and theoretical tractability. As practitioners increasingly share or reuse post-training datasets across models and platforms, mitigating sensitivity becomes crucial.

**Potential failure modes.** Although our lower-bound constructions are stylized for analytic clarity, they highlight patterns that may emerge in practice — for instance, systematic bias arises when comparisons frequently involve actions suppressed by the reference policy. Our theoretical framework provides a principled way to study such effects and to identify the structural features of data distributions that most influence worst-case misalignment.

Finally, the distortion framework opens up many more questions: How large is the distortion of alignment methods besides RLHF, DPO, and NLHF? Can we extend the model to capture the generalization of preferences across states? Can distortion be reduced with a little additional information? And can we go beyond average utility to capture finer-grained notions? After all, high average utility is necessary, but not sufficient, for successful alignment to a heterogeneous population.

---

[11]The finite-sample bounds even improve in the latter model since each pair appears only once.

## Acknowledgments and Disclosure of Funding

We thank Mark Bedaywi, Jim Dai, Sonja Kraiczy, Soroosh Shafiee, and Eric Zhao for helpful conversations. This work was supported in part by the National Science Foundation under grant CCF-2145898, by the Office of Naval Research under grant N00014-24-1-2159, an Alfred P. Sloan fellowship, and a Schmidt Sciences AI2050 fellowship. Part of this work was performed while P.G. was at the Simons Institute for the Theory of Computing as a FODSI research fellow, for which he acknowledges the NSF's support through grant DMS-2023505.

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

# A Related work

**Reward-based and reward-free alignment methods.** The RLHF pipeline typically includes first training a reward model via maximum likelihood estimation (MLE), then applying RL algorithms such as Proximal Policy Optimization (PPO) [44] to optimize a policy that maximizes the reward [59, 7]. Rafailov et al. [43] proposes an alternative approach, Direct Preference Optimization (DPO), which bypasses explicit reward model training by directly optimizing an equivalent objective derived from the closed form of KL-constrained reward-maximizing policy. While the original formulation is based on a single Bradley–Terry model, we show in Appendix F.4 that the equivalence extends to settings with heterogeneous preferences. Building on the DPO framework, several recent methods including $\chi$PO [30], RPO [32] and SimPO [34] have been proposed to improve the robustness and effectiveness.

Azar et al. [5] introduce $\Psi$PO, another reward-free method that optimizes the expectation of a $\Psi$-transformation of the win-rates estimated from the offline comparison data. When $\Psi$ is the identity function, the resulting method — IPO — reduces to directly optimizing the normalized Borda count. Since RLHF is implicitly optimizing the normalized Borda count [46, 42], this connection implies that IPO, DPO, and RLHF are all equivalent in the unregularized/unconstrained setting.

Another reward-free method is Nash Learning from Human Feedback (NLHF) [36] and its variants [48, 52, 11], which finds the Nash equilibrium of a game defined over the win-rate margins (i.e., $p(x \succ y) - p(y \succ x)$) via online learning or self-play style algorithms. Maura-Rivero et al. [33] point out that NLHF can be viewed as a natural generalization of the Maximal Lotteries rule in social choice. Wang et al. [50] consider finding the Nash equilibrium of the win-rate matrix and reduce the problem to multiagent reward-based RL. They provide an impossibility result, showing the optimal policy is indeterminate when the underlying ranking model (e.g., Bradley-Terry with certain temperature) is unknown. In contrast, our results show that even when the ranking model is known, the optimal policy can remain nonidentifiable due to preference heterogeneity.

**AI Alignment under heterogeneous user preferences.** A growing body of recent works studies algorithms for AI alignment under heterogeneous user preferences. Siththaranjan et al. [46] points out that RLHF implicitly optimizes the normalized Borda count, which can lead to poor outcomes in the social choice setting. To address this, they propose Distributional Preference Learning (DPL), a method that estimates a distribution of score values for each alternative. Another line of work deals with heterogeneity by clustering user preferences and learning several reward models at once, then aggregate the learned reward models using various techniques such as max-min optimization, which optimizes the worst-case reward among all clusters [17, 13], or through aggregation rules motivated by axiomatic properties in social choice theory [58, 39]. Poddar et al. [40] proposes a variational inference approach that infers user-specific latent variables from preference data which enables steerable personalized language models. Chen et al. [15] proposes a framework based on the ideal point model, which learns a latent space of user preferences that can few-shot generalize to unseen users.

**Statistical and Axiomatic Perspectives on Preference Aggregation.** Maximum likelihood estimators (MLE), which serves as the core of the widely-used RLHF pipeline, can be viewed as voting rules: given a set of rankings, they output a score for each alternative, thereby producing a single aggregated ranking. This connection was first observed by Conitzer and Sandholm [19], who show that any scoring-based voting rule is a maximum likelihood estimator under a specific noise model. A rich literature in social choice theory has studied the axiomatic properties of such MLE-based voting rules under various randomized ranking models [6, 53, 37, 27, 42]. Notably, Ge et al. [27] analyzes the axiomatic properties of MLE-based AI alignment methods under the Bradley-Terry model for linear utility functions.

On the learning side, several works study the problem of learning mixture models from ranking data, see textbooks [2, 54] for a comprehensive overview. Recently, Wang et al. [51], Tatli et al. [49] focus on learning metric spaces from pairwise preferences. Our work is notably related to the results on the non-identifiability of learning mixture of Bradley-Terry models from pairwise or $k$-wise preferences [57, 56, 55]. We build on these results to quantify the loss of utility due to non-identifiability by proving a voting-rule independent distortion lower bound.

**Distortion of randomized voting and RLHF.** The framework of implicit utilitarian voting, i.e., of comparing voting rules in terms of their distortion was introduced by Procaccia and Rosenschein [41], which has since sparked a large body of work — both in the original utility setting [12, 10, 23, 26, 9, 23] and in the metric setting [3, 4, 28, 14, 31]. Several recent works have highlighted the importance of using distortion as a metric to evaluate the quality of AI alignment methods. Dai and Fleisig [21] draw a conceptual connection between social choice and RLHF, and propose to apply the notion of distortion to RLHF. Goyal and Sarmasarkar [29] uses alignment as motivation for studying the metric distortion of probabilistic voting rules under Bradley-Terry and other random utility models, where the voters and candidates are assumed to lie in a common metric space satisfying triangle inequality. We not only study the non-metric distortion (which is more expressive), but also go beyond the social choice setting to consider the alignment setting in which output policies are constrained to remain close to a given reference policy. More broadly, our work also contributes to the growing line of research on the intersection of social choice theory and RLHF, as advocated in recent position papers [20, 35].

# B  Distortion in AI Leaderboards

The AI leaderboard setting fits natural into our social-choice model, where $\mathcal{D}$ captures a random user's utility for the responses of different models to a random prompt (drawn from an arbitrary joint distribution over users and prompts), and AvgUtil quantified the average utility a model delivers for a random user and task, which we call the model's *usability*.

Since Chatbot Arena and RLHF are based on the same MLE, our distortion bounds on RHLF in the social choice setting imply that the usability of the top-ranked language model (i.e., the Borda winner) may be $(1 - o(1)) \cdot \beta$ times worse than the usability of some other ranked model (Theorem 5) (but at most by a $O(\beta^2)$ factor, see Theorem 2). Our results in an extended setting in which comparison pairs are drawn in a correlated way (Section 5) show that Chatbot Arena's ranking is highly sensitive to the distribution of LLM pairs. For certain correlated distributions, the gap in usability could be unbounded (Theorem 9), which is concerning since Chatbot Arena adaptively oversamples new and highly ranked models.

These findings suggest that current leaderboard rankings may not fully reflect true model quality. Could alternative aggregation rules, such as Maximal Lotteries or the Copeland voting rule, provide more accurate assessments of model usability and be more robust to the choice of sampling distribution? Does adaptive sampling introduce systematic biases that exacerbate the distortion of current pipelines? Addressing these questions is an important direction for future work to ensure the fidelity of leaderboard-based evaluations.

# C  Linearization Lemma for Expected Win-Rates

**Lemma 1** (Linearization of Expected Win-Rates). *Let $L := \sigma'(0) = 1/4$ and $\ell_\beta := \frac{\sigma(\beta) - \frac{1}{2}}{\beta} = \frac{1}{2\beta} \cdot \frac{1 - e^{-\beta}}{1 + e^{-\beta}}$. For any pair of alternatives $x, y \in A$, we have*

$$\beta \cdot (\ell_\beta \cdot \mathsf{AvgUtil}(x) - L \cdot \mathsf{AvgUtil}(y)) \leq p(x \succ y) - \tfrac{1}{2} \leq \beta \cdot (L \cdot \mathsf{AvgUtil}(x) - \ell_\beta \cdot \mathsf{AvgUtil}(y)).$$

*Proof of Lemma 1.* We prove this lemma by linearizing the sigmoid function $\sigma(z) = \frac{1}{1 + e^{-z}}$ in the domain of $z \in [-\beta, \beta]$. When $z \in [0, \beta]$, the sigmoid function is concave and increasing, thus we have $\sigma(z) \leq \sigma'(0) \cdot z + \sigma(0) = \frac{1}{2} + Lz$, where $L = \frac{1}{4}$ is the derivative $\sigma'(0)$. When $z \in [-\beta, 0]$, the sigmoid function is convex, thus we have $\sigma(z) \leq \left(1 + \frac{z}{\beta}\right) \sigma(0) - \frac{z}{\beta} \sigma(-\beta) = \frac{1}{2} + l_\beta \cdot z$, where $l_\beta = \frac{\sigma(\beta) - \frac{1}{2}}{\beta}$ is the slope of the line connecting $(-\beta, \sigma(-\beta))$ and $(0, \sigma(0))$.

Plugging the above bounds into $\sigma\Big(\beta \cdot (u(x) - u(y))\Big)$, we have that

$$
\begin{aligned}
\sigma\Big(\beta \cdot (u(x) - u(y))\Big) - \frac{1}{2} &\leq \beta \cdot (u(x) - u(y)) \cdot \Big(L \cdot \mathbb{1}_{u(x) - u(y) \geq 0} + l_\beta \cdot \mathbb{1}_{u(x) - u(y) < 0}\Big) \\
&\leq \beta \cdot (L \cdot u(x) - l_\beta \cdot u(y)).
\end{aligned}
$$

Finally, taking an expectation over $u \sim \mathcal{D}$, we have that

$$p(x \succ y) - \frac{1}{2} \leq \beta \left( L \cdot \mathop{\mathbb{E}}_{u \sim \mathcal{D}} [u(x)] - l_\beta \cdot \mathop{\mathbb{E}}_{u \sim \mathcal{D}} [u(y)] \right) = \beta \left( L \cdot \mathsf{AvgUtil}(x) - l_\beta \cdot \mathsf{AvgUtil}(y) \right).$$

This completes the proof of the upper bound. The lower bound follows from applying the same argument to $p(y \succ x)$ and using the fact that $p(x \succ y) = 1 - p(y \succ x)$. □

# D  Finite-Sample Convergence Bounds

In this section, we use standard concentration techniques to derive finite-sample convergence bounds for the normalized Borda score and the empirical win rate. The lemmas presented in this section will serve as a building block for proving finite-sample guarantees for the alignment methods studied in Sections 3 and 4.

## D.1  Estimation Error of Win-Rates

**Lemma 10.** *For any instance $\mathcal{D}$ with any number of alternatives $m$, any distribution $\mu$ over alternatives with $\mu_{\min} = \min_{x \in A} \mu(x)$, and $n$ i.i.d. users sampled from $\mathcal{D}$ where each user labels $d$ comparison pairs following the Bradley-Terry model with temperature $\beta$, we have that with probability at least $1 - \delta$ where $\delta \geq m^2 \exp \left( -\frac{nd\mu_{\min}^2}{8} \right)$, the empirical win rates $p_n(x \succ y) := \frac{\#(x \succ y)}{\#(x \succ y) + \#(y \succ x)}$ satisfies that:*

$$\forall x, y \in A, \quad |p_n(x \succ y) - p(x \succ y)| \leq O \left( \sqrt{\frac{\log(m/\delta)}{n \cdot \min\{1, d \cdot \mu_{\min}^2\}}} + \frac{\log(m/\delta)}{n\mu_{\min}^2} \right).$$

*Proof of Lemma 10.* We first bound the estimation error of $p_n(x \succ y)$ for a fixed pair $x, y \in \binom{A}{2}$. Here we assume $x \neq y$ without loss of generality, because the estimation error for the $x = y$ case is 0.

Since each voter $i \in [n]$ is asked to label $d$ pairwise comparisons, if each of them are asked to label a pair $\{x, y\}$ multiple times, their answer will be consistent. Therefore, we can equivalently rewrite the process of sampling $p_n(x \succ y)$ as follows:

1. Draw $k_1, \ldots, k_n \stackrel{\text{i.i.d.}}{\sim} \mathrm{Binomial}(d, q)$ to represent the number of times the $i$-th voter is asked to label $\{x, y\}$, where $q := 2\mu(x)\mu(y)$ is the probability that each comparison pair is $\{x, y\}$;

2. Draw $p_1, \ldots, p_n \stackrel{\text{i.i.d.}}{\sim} \mathrm{Bernoulli}(p)$ to represent the preference of the $i$-th voter on pair $\{x, y\}$, where $p := p(x \succ y)$ is the probability that a fresh voter prefers $x$ over $y$. In particular, we have $p_i \perp k_i$ because the sampling of voters and comparison pairs are independent;

3. Each voter $i \in [n]$ contributes $k_i \cdot p_i$ to $\#(x \succ y)$ and $k_i \cdot (1 - p_i)$ to $\#(y \succ x)$.

As a result, the empirical win rate $p_n(x \succ y)$ can be rewritten as:

$$p_n(x \succ y) = \frac{\#(x \succ y)}{\#(x \succ y) + \#(y \succ x)} = \frac{\sum_{i=1}^n k_i p_i}{\sum_{i=1}^n k_i}.$$

The error term is then given by:

$$p_n(x \succ y) - p(x \succ y) = \frac{\sum_{i=1}^n k_i p_i}{\sum_{i=1}^n k_i} - p = \frac{\sum_{i=1}^n k_i (p_i - p)}{\sum_{i=1}^n k_i}.$$

Now we use Bernstein's inequality to bound the numerator. We start by bounding the variance of random variable $Z_i := k_i(p_i - p)$. Note that $\mathbb{E}[Z_i] = \mathbb{E}[k_i] \cdot \mathbb{E}[p_i - p] = 0$ because $k_i$ and $p_i - p$ are independent. Therefore, we have

$$\mathrm{Var}(Z_i) = \mathbb{E}[Z_i^2] = \mathbb{E}[k_i^2] \cdot \mathbb{E}[(p_i - p)^2] \leq \mathbb{E}[k_i^2] = \mathrm{Var}(k_i) + (\mathbb{E}[k_i])^2 = dq(1 - q) + d^2 q^2.$$

According to Bernstein's inequality, we have that with probability at least $1 - \delta$,

$$\left| \sum_{i=1}^{n} Z_i \right| = \left| \sum_{i=1}^{n} k_i(p_i - p) \right| \leq \sqrt{2n(dq(1-q) + d^2 q^2) \log(2/\delta)} + 3d \log(2/\delta). \quad (3)$$

Now we bound the denominator. Note that $\mathbb{E}\left[k_i\right] = dq$ and $\mathrm{Var}\left(k_i\right) = dq(1-q)$. From the Chernoff bound, we have that with probability at least $1 - e^{-\frac{ndq}{8}}$,

$$\sum_{i=1}^{n} k_i \geq \frac{n \mathbb{E}\left[k_i\right]}{2} = \frac{ndq}{2}. \quad (4)$$

Combining the bounds in Equation (3) and Equation (4), we have that when $\delta \geq e^{-\frac{ndq}{8}}$, with probability at least $1 - 2\delta$, for a fixed pair $x, y \in A$, we have

$$|p_n(x \succ y) - p(x \succ y)| \leq \frac{\sqrt{2n(dq(1-q) + d^2 q^2) \log(2/\delta)} + 3d \log(2/\delta)}{ndq/2}$$

$$\leq O\left( \sqrt{\frac{(1 - q + dq) \log(1/\delta)}{ndq}} + \frac{\log(1/\delta)}{nq} \right)$$

where we use the fact that $\frac{1-q+dq}{dq} \leq \frac{2}{\min\{1, dq\}}$ and $q = 2\mu(x)\mu(y) \geq \mu_{\min}^2$ to obtain:

$$\leq O\left( \sqrt{\frac{\log(1/\delta)}{n \min\{1, d \cdot \mu_{\min}^2\}}} + \frac{\log(1/\delta)}{n\mu_{\min}^2} \right).$$

Finally, by union bound over all $\binom{m}{2}$ pairs, we have that with probability at least $1 - \delta$ where $\delta \geq m^2 \exp\left(-\frac{nd\mu_{\min}^2}{8}\right)$, the following holds simultaneously for all $x, y \in A$:

$$|p_n(x \succ y) - p(x \succ y)| \leq O\left( \sqrt{\frac{\log(m/\delta)}{n \cdot \min\{1, d \cdot \mu_{\min}^2\}}} + \frac{\log(m/\delta)}{n\mu_{\min}^2} \right).$$

The proof is complete. $\qquad \square$

## D.2 Estimation Error of Normalized Borda Score

**Lemma 11.** *For any instance $\mathcal{D}$ with any number of alternatives $m$, any distribution $\mu$ over alternatives with $\mu_{\min} = \min_{x \in A} \mu(x)$, and $n$ i.i.d. users sampled from $\mathcal{D}$ where each user labels $d$ comparison pairs, the normalized Borda score $\mathsf{BC}_n(x)$ of any alternative $x \in A$ satisfies that with probability at least $1 - \delta$ where $\delta \geq 2m \exp(-\frac{nd\mu_{\min}}{8})$,*

$$\forall x \in A, \quad |\mathsf{BC}_n(x) - \mathsf{BC}^\star(x)| \leq O\left( \sqrt{\frac{\log(m/\delta)}{n \cdot \min\{1, d \cdot \mu_{\min}^2\}}} + \frac{m \log(m/\delta)}{n\mu_{\min}} \right), \quad (5)$$

*where $\mathsf{BC}^\star(x)$ is the limiting normalized Borda score of candidate $x$, defined as*

$$\mathsf{BC}^\star(x) := \sum_{y \in A} \mu(y) \cdot p(x \succ y) = \frac{1}{2}\mu(x) + \sum_{y \neq x} \mu(y) \cdot p(x \succ y). \quad (6)$$

*Proof.* We first bound the estimation error $|\mathsf{BC}_n(x) - \mathsf{BC}^\star(x)|$ for a fixed alternative $x \in A$. For notational simplicity, we use $T_n(x)$ to denote the number of comparison pairs involving $x$, and $W_n(x)$ to denote the number of comparison pairs where $x$ is the winner, i.e.,

$$T_n(x) = 2\#(x \succ x) + \sum_{y \neq x} \#(x \succ y) + \#(y \succ x), \quad W_n(x) = \#(x \succ x) + \sum_{y \neq x} \#(x \succ y).$$

The normalized Borda score of $x$ is then given by $\mathsf{BC}_n(x) = \frac{W_n(x)}{T_n(x)}$. It is then easy to see that

$$\mathbb{E}\left[T_n(x)\right] = nd\left(2\mu(x)^2 + \sum_{y \neq x} 2\mu(x)\mu(y)\right) = 2nd\mu(x);$$

$$\mathbb{E}\left[W_n(x)\right] = nd\left(\mu(x)^2 + \sum_{y \neq x} \mu(x)\mu(y)p(x \succ y)\right) = nd\mu(x)\left(\frac{1}{2}\mu(x) + \sum_{y \neq x}\mu(y)p(x \succ y)\right).$$

The limiting Borda score $\mathsf{BC}^\star(x)$ is then given by the ratio of the above two expectations, i.e.,

$$\mathsf{BC}^\star(x) = \frac{\mathbb{E}\left[W_n(x)\right]}{\mathbb{E}\left[T_n(x)\right]} = \frac{1}{2}\mu(x) + \sum_{y \neq x}\mu(y)p(x \succ y).$$

We can thus decompose the estimation error $|\mathsf{BC}_n(x) - \mathsf{BC}^\star(x)|$ as follows:

$$\begin{aligned}
|\mathsf{BC}_n(x) - \mathsf{BC}^\star(x)| &= \left|\frac{W_n(x)}{T_n(x)} - \frac{\mathbb{E}\left[W_n(x)\right]}{\mathbb{E}\left[T_n(x)\right]}\right| \\
&\leq \frac{|W_n(x) - \mathbb{E}\left[W_n(x)\right]|}{T_n(x)} + \underbrace{\frac{\mathbb{E}\left[W_n(x)\right]}{\mathbb{E}\left[T_n(x)\right]}}_{\mathsf{BC}^\star(x) \leq 1} \cdot \frac{|T_n(x) - \mathbb{E}\left[T_n(x)\right]|}{T_n(x)} \\
&\leq \frac{|W_n(x) - \mathbb{E}\left[W_n(x)\right]|}{T_n(x)} + \frac{|T_n(x) - \mathbb{E}\left[T_n(x)\right]|}{T_n(x)}.
\end{aligned}$$

Now we bound the two terms $|W_n(x) - \mathbb{E}\left[W_n(x)\right]|$ and $|T_n(x) - \mathbb{E}\left[T_n(x)\right]|$ separately, and apply the union bound at the end.

**(I). Bounding $|W_n(x) - \mathbb{E}\left[W_n(x)\right]|$:** For each $y \neq x$, we can write bound the deviation $|\#(x \succ y) - \mathbb{E}\left[\#(x \succ y)\right]|$ using the same argument as in the proof of Lemma 10. Specifically, we can write $\#(x \succ y)$ as a sum of i.i.d. random variables $k_i p_i$ where $k_i \sim \text{Binomial}(d, q_{x,y})$ and $p_i \sim \text{Bernoulli}(p(x \succ y))$, where $q_{x,y} = 2\mu(x)\mu(y)$ is the probability that each comparison pair is $\{x, y\}$. Therefore, we have

$$\begin{aligned}
\text{Var}\left(k_i p_i\right) &= \text{Var}\left(k_i\right) \cdot \text{Var}\left(p_i\right) + \text{Var}\left(k_i\right)\mathbb{E}\left[p_i^2\right] + \text{Var}\left(p_i\right)\mathbb{E}\left[k_i\right]^2 \\
&\leq 2dq_{x,y}(1 - q_{x,y} + dq_{x,y})
\end{aligned}$$

since $1 - q_{x,y} + dq_{x,y} \leq \frac{2dq_{x,y}}{\min\{1, dq_{x,y}\}} \leq \frac{2dq_{x,y}}{\min\{1, d\mu_{\min}^2\}}$, we can further bound the variance as

$$\leq \frac{2(dq_{x,y})^2}{\min\{1, d\mu_{\min}^2\}} = \frac{8(d\mu(x)\mu(y))^2}{\min\{1, d\mu_{\min}^2\}}.$$

Thus, by Bernstein's inequality, with probability at least $1 - \delta'$,

$$\left|\#(x \succ y) - \mathbb{E}\left[\#(x \succ y)\right]\right| \leq 4d\mu(x)\mu(y)\sqrt{\frac{n\log(2/\delta')}{\min\{1, d \cdot \mu_{\min}^2\}}} + 3d\log(2/\delta').$$

On the other hand, for the comparison of $x$ with itself, we have that $\#(x \succ x) \sim \text{Binomial}(nd, \mu(x)^2)$. Therefore, with probability at least $1 - \delta'$,

$$\begin{aligned}
\left|\#(x \succ x) - \mathbb{E}\left[\#(x \succ x)\right]\right| &\leq \sqrt{2nd\mu(x)^2\log(2/\delta')} + 3d\log(2/\delta') \\
&\leq 4d\mu(x)^2\sqrt{\frac{n\log(2/\delta')}{\min\{1, d \cdot \mu_{\min}^2\}}} + 3d\log(2/\delta').
\end{aligned}$$

Applying a union bound over all the $m$ alternatives $y \in A$, we have that with probability at least $1 - m\delta'$,

$$|W_n(x) - \mathbb{E}\left[W_n(x)\right]| \leq \sum_{y \neq x} \left| \#(x \succ y) - \mathbb{E}\left[\#(x \succ y)\right]\right|$$

$$\leq \sum_{y \neq x} \left( 4d\mu(x)\mu(y)\sqrt{\frac{n \log(2/\delta')}{\min\{1, d \cdot \mu_{\min}^2\}}} + 3d \log(2/\delta') \right)$$

$$\leq 4d\mu(x)\sqrt{\frac{n \log(2/\delta')}{\min\{1, d \cdot \mu_{\min}^2\}}} + 3md \log(2/\delta').$$

**(II). Bounding $|T_n(x) - \mathbb{E}\left[T_n(x)\right]|$:**   We can write $T_n(x)$ as a sum of i.i.d. random variables:

$$T_n = \sum_{i=1}^{n} \sum_{j=1}^{d} (\mathbb{1}_{x_i^j = x} + \mathbb{1}_{y_i^j = x})$$

Since each comparison pair $x_i^j, y_i^j$ is sampled independently from $\mu \times \mu$, we have that $T_n \sim$ Binomial$(2nd, \mu(x))$. Therefore, with probability at least $1 - \delta'$,

$$\left| T_n(x) - \mathbb{E}\left[T_n(x)\right] \right| \leq 2\sqrt{nd\mu(x) \log(2/\delta')} + 3 \log(2/\delta').$$

In addition, with probability at least $1 - \exp(-\frac{nd\mu(x)}{4})$, we also have

$$T_n \geq \frac{\mathbb{E}\left[T_n\right]}{2} = nd\mu(x).$$

**(III). Combining the two bounds:**   Finally, combining the above bounds on $|W_n(x) - \mathbb{E}\left[W_n(x)\right]|$ and $|T_n(x) - \mathbb{E}\left[T_n(x)\right]|$, together with the bound on the denominator $T_n(x)$, we have that with probability at least $1 - 2m\delta' - \exp(-\frac{nd\mu(x)}{4})$,

$$|\mathsf{BC}_n(x) - \mathsf{BC}^\star(x)| \leq \frac{|W_n(x) - \mathbb{E}\left[W_n(x)\right]|}{T_n(x)} + \frac{|T_n(x) - \mathbb{E}\left[T_n(x)\right]|}{T_n(x)}$$

$$\lesssim \frac{1}{nd\mu(x)} \left( d\mu(x)\sqrt{\frac{n \log(1/\delta')}{\min\{1, d \cdot \mu_{\min}^2\}}} + md \log(1/\delta') + \sqrt{nd\mu(x) \log(2/\delta')} \right)$$

$$\lesssim \sqrt{\frac{\log(1/\delta')}{n \cdot \min\{1, d \cdot \mu_{\min}^2\}}} + \frac{m \log(1/\delta')}{n\mu_{\min}}.$$

Finally, setting $\delta' = \frac{\delta}{4m^2}$ and taking a union bound over all the $m$ alternatives $x \in A$, we have that when $\delta \geq 2m \exp(-\frac{nd\mu_{\min}}{8})$, with probability at least $1 - \delta$, the above bound holds simultaneously for all $x \in A$. This completes the proof. $\qquad \square$

# E   Supplemental Materials for Section 3

## E.1   Upper Bound for Borda

**Theorem 2** (Borda Distortion Upper Bound). *For any instance $\mathcal{D}$ with any number of alternatives $m$, distribution $\mu$ over alternatives, and temperature $\beta$, Borda has at most distortion $\left(\frac{\beta}{2} \cdot \frac{1+e^{-\beta}}{1-e^{-\beta}}\right)^2 = O(\beta^2)$. In the finite-sample regime, we have that*

$$\mathsf{AvgUtil}_n(\mathsf{Borda}) \geq \left(\frac{2}{\beta} \cdot \frac{1-e^{-\beta}}{1+e^{-\beta}}\right)^2 \cdot \max_{x^\star \in A} \mathsf{AvgUtil}(x^\star) - O\left(\frac{1}{\beta^2}\sqrt{\frac{\log(mn\beta)}{n \cdot \min\{1, d\mu_{\min}^2\}}} + \frac{m \log(mn\beta)}{n \cdot \beta^2 \mu_{\min}^2}\right).$$

*Proof of Theorem 2.* From Lemma 11, we have that with probability at least $1 - \delta$, all $x \in A$ satisfy that $|\mathsf{BC}_n(x) - \mathsf{BC}^\star(x)| \leq \varepsilon_{n,d}(\delta)$, where

$$\varepsilon_{n,d}(\delta) = O\left(\sqrt{\frac{\log(m/\delta)}{n \cdot \min\{1, d \cdot \mu_{\min}^2\}}} + \frac{m \log(m/\delta)}{n\mu_{\min}}\right).$$

Following the proof sketch in Section 3.1, we use $\hat{x} = \text{argmax}_{x \in A}\, \mathsf{BC}_n(x)$ to denote the Borda winner, and $x^\star = \text{argmax}_{x \in A}\, \mathsf{AvgUtil}(x)$ to denote the true welfare maximizer.

Since $\mathsf{BC}_n(\hat{x}) \geq \mathsf{BC}_n(x^\star)$, we have

$$\mathsf{BC}^\star(\hat{x}) - \mathsf{BC}^\star(x^\star) \geq -2\varepsilon_{n,d}(\delta). \tag{7}$$

For the limiting Borda score $\mathsf{BC}^\star(x)$, the argument in Section 3.1 shows that

$$\mathsf{BC}^\star(\hat{x}) - \mathsf{BC}^\star(x^\star) \leq \beta \cdot (L \cdot \mathsf{AvgUtil}(\hat{x}) - \ell_\beta \cdot \mathsf{AvgUtil}(x^\star) + (L - \ell_\beta) \cdot \mathsf{AvgUtil}(\mu))$$
$$\leq \beta \cdot \left( \frac{L^2}{\ell_\beta} \cdot \mathsf{AvgUtil}(\hat{x}) - \ell_\beta \cdot \mathsf{AvgUtil}(x^\star) \right) \tag{8}$$

Therefore, Combining Equations (7) and (8), we have

$$-2\varepsilon_{n,d}(\delta) \leq \beta \cdot \left( \frac{L^2}{\ell_\beta} \cdot \mathsf{AvgUtil}(\hat{x}) - \ell_\beta \cdot \mathsf{AvgUtil}(x^\star) \right) \;\Rightarrow\; \mathsf{AvgUtil}(\hat{x}) \geq \left( \frac{\ell_\beta}{L} \right)^2 \mathsf{AvgUtil}(x^\star) - \frac{2\ell_\beta \cdot \varepsilon_{n,d}(\delta)}{\beta L^2}.$$

Combining this with the failure probability $\delta$ of the above argument, the expected social welfare $\mathsf{AvgUtil}_n(\text{Borda})$ satisfies that

$$\mathsf{AvgUtil}_n(\text{Borda}) \geq (1 - \delta) \cdot \left( \left( \frac{\ell_\beta}{L} \right)^2 \mathsf{AvgUtil}(x^\star) - \frac{2\ell_\beta \cdot \varepsilon_{n,d}(\delta)}{\beta L^2} \right)$$
$$\geq \left( \frac{\ell_\beta}{L} \right)^2 \mathsf{AvgUtil}(x^\star) - O \left( \frac{\varepsilon_{n,d}(\delta)}{\beta} \cdot \left( \frac{\ell_\beta}{L} \right) + \delta \cdot \left( \frac{\ell_\beta}{L} \right)^2 \right).$$

Finally, we set the failure probability to be

$$\delta = \Theta \left( \frac{L}{\beta \cdot \ell_\beta} \cdot \sqrt{\frac{1}{n \cdot \min\{1, d \cdot \mu_{\min}^2\}}} \right)$$

(which satisfies the condition in Lemma 11 for large $n$), we have

$$\mathsf{AvgUtil}_n(\text{Borda}) \geq \left( \frac{\ell_\beta}{L} \right)^2 \mathsf{AvgUtil}(x^\star) - O \left( \frac{1}{\beta^2} \sqrt{\frac{\log(mn\beta)}{n \cdot \min\{1, d\mu_{\min}^2\}}} + \frac{m \log(mn\beta)}{n \cdot \beta^2 \mu_{\min}^2} \right),$$

which completes the proof. $\qquad \square$

## E.2 Lower Bound for Borda

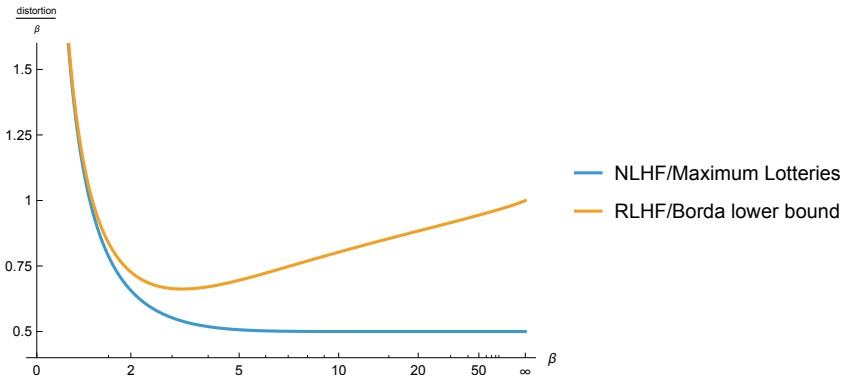

Figure 3: Comparison of the distortion achieved by NLHF/Maximum Lotteries and the lower bound on RLHF/Borda in Theorem 5, both as a fraction of $\beta$. The figure illustrates that NLHF has a worse distortion for every value of $\beta > 0$ (for worst-case distributions $\mu$); in particular, the distortion of RLHF for large $\beta$ is at least $\beta - o(\beta)$, whereas the distortion of NLHF is $\beta/2 + o(\beta)$.

**Theorem 12** (Lower Bound for Borda; Formal Version of Theorem 5). *For any $\beta > 0$ and $m \geq 3$, the Borda voting rule (and, hence, RHLF) cannot guarantee a distortion better than $\max_{0 < \gamma < 1} \frac{\beta}{2} \frac{1 + e^{-\beta}}{1 - e^{-\beta}} \cdot \left(1 - \gamma + \frac{\sigma(\beta\gamma) - 1/2}{\sigma(\beta) - 1/2}\right)$. This bound is strictly higher than the voting-rule independent lower bound $\frac{\beta}{2} \frac{1 + e^{-\beta}}{1 - e^{-\beta}}$ for all $\beta$ and is at least $(1 - o(1)) \beta$ as $\beta \to \infty$.*

*Proof of Theorem 12.* Without loss of generality, we may assume that $m = 3$. If $m > 3$, we can repeatedly "split" some alternative $x$ in two new alternatives $y, y'$ (where each user has the utility for $y, y'$ as for the original alternative $x$, and $\mu(y) + \mu(y')$ is equal to the original mass of $x$ in $\mu$). In this operation, the social welfares and Borda scores of $y, y'$ in the new instance are equal to the social welfare and Borda score of $x$ in the original instance, and the social welfares and Borda scores of all other alternatives do not change.

For any $0 < \epsilon < 1$, $0 \leq \epsilon' < 1 - \epsilon$, and $0 < \gamma < 1$, consider the following distribution $\mathcal{D}$ of utilities over alternatives $(a, b, c)$:

$$(u(a), u(b), u(c)) = \begin{cases} (1 - \gamma, 1, 0) & \text{with probability } p_A := \frac{\sigma(\beta\epsilon) - 1/2}{\sigma(\beta) + \sigma(\beta\epsilon) - 1} \\ (1, 0, \epsilon) & \text{with probability } p_B := p_A \cdot \frac{\sigma(\beta\gamma) - 1/2}{\sigma(\beta) - 1/2} \\ (0, 0, \epsilon + \epsilon') & \text{with probability } 1 - p_A - p_B. \end{cases}$$

One verifies that $0 < p_A, p_B$ and $p_A + p_B < 1$, so this describes a valid probability distribution for all $\epsilon, \epsilon', \gamma$ and each type of utilities has positive probability of being drawn. Assuming that $\epsilon' = 0$, it must be true that $p(b \succ c) = 1/2 = p(c \succ b)$ because

$$p_A \cdot \sigma(\beta(1 - 0)) + (1 - p_A) \cdot \sigma(\beta(0 - \epsilon)) = p_A \cdot \left(\sigma(\beta) - \underbrace{\sigma(-\beta\epsilon)}_{=1 - \sigma(\beta\epsilon)}\right) + \underbrace{\sigma(-\beta\epsilon)}_{=1 - \sigma(\beta\epsilon)}$$

$$= p_A \cdot \left(\sigma(\beta) + \sigma(\beta\epsilon) - 1\right) + 1 - \sigma(\beta\epsilon)$$

$$= \sigma(\beta\epsilon) - 1/2 + 1 - \sigma(\beta\epsilon) = 1/2.$$

If $\epsilon' > 0$, it must be the case that $p(c \succ b) > 1/2$ by monotonicity. A similar chain of algebra shows that $p(a \succ b) = 1/2 = p(b \succ a)$:

$$p_A \cdot \sigma(-\beta\gamma) + p_B \cdot \sigma(\beta) + \frac{1 - p_A - p_B}{2} = p_A \cdot \underbrace{\left(\sigma(-\beta\gamma) - 1/2\right)}_{=1/2 - \sigma(\beta\gamma)} + p_B \cdot \underbrace{\left(\sigma(\beta) - 1/2\right)}_{=p_A \cdot (\sigma(\beta\gamma) - 1/2)} + 1/2 = 1/2.$$

For any $\epsilon, \gamma$ and positive $\epsilon'$, note that, as the number of samples goes to infinity, the Borda score of the alternatives concentrate around their expected values:

$$\mathsf{BC}(a) \to \frac{1}{2}\mu(a) + \frac{1}{2}\mu(b) + p(a \succ c)\,\mu(c)$$

$$\mathsf{BC}(b) \to \frac{1}{2}\mu(a) + \frac{1}{2}\mu(b) + p(b \succ c)\,\mu(c)$$

$$\mathsf{BC}(c) \to p(c \succ a)\,\mu(a) + p(c \succ b)\,\mu(b) + \frac{1}{2}\mu(c).$$

Recall that $p(c \succ b) > 1/2 > p(b \succ c)$. Regardless of what $p(a \succ c) = 1 - p(c \succ a)$ may be, for any distribution $\mu$ with small enough $\mu(a), \mu(c)$ (and hence large $\mu(b)$), the expected Borda score of $c$ will be strictly larger than that of $a$ and $b$. By concentration, for large enough $n$, the Borda voting rule will almost surely select $c$ as the winner, and the Borda's distortion for that $\mu$ will be at least

$$\frac{\max_{x \in A} \mathsf{AvgUtil}(x)}{\mathsf{AvgUtil}(c)} \geq \frac{\mathsf{AvgUtil}(a)}{\mathsf{AvgUtil}(c)} = \frac{p_A (1 - \gamma) + p_B}{p_B \epsilon + (1 - p_A - p_B)(\epsilon + \epsilon')}. \tag{9}$$

We can now derive lower bounds on the distortion of Borda by defining sequences of parameters $\epsilon, \epsilon', \gamma$ (and implicitly, a sequence of corresponding distributions $\mu$), and considering the limit of Eq. (9). In each such sequence, we treat $\gamma$ as a fixed parameter, but let $\epsilon' := \epsilon^2$ and letting $\epsilon$ go to 0. As $\epsilon \to 0$, it holds that $p_A \to 0$ (because its numerator $\sigma(\beta\epsilon) - 1/2 \to 1/2 - 1/2 = 0$), that $p_B \to 0$ (since it is a constant multiple of $p_A$), and hence both the numerator and denominator of

Eq. (9) converge to 0. We apply l'Hôpital's rule to determine the limit. Treating $p_A$ and $p_B$, as well as the numerator $num$ and denominator $den$ of the equation as functions in $\epsilon$, we observe that

$$p'_A(0) = \frac{\beta}{4 \cdot (\sigma(\beta) - 1/2)}$$

$$p'_B(0) = \frac{\beta \cdot (\sigma(\beta\gamma) - 1/2)}{4 \cdot (\sigma(\beta) - 1/2)^2}$$

$$num'(0) = \frac{\beta}{4 \cdot (\sigma(\beta) - 1/2)} \cdot \left(1 - \gamma + \frac{\sigma(\beta\gamma) - 1/2}{\sigma(\beta) - 1/2}\right)$$

$$den'(0) = \underbrace{p_B(0)}_{=0} \cdot 1 + p'_B(0) \cdot 0 + \underbrace{(1 - p_A(0) - p_B(0))}_{=1} \cdot (1 + 2 \cdot 0) + 0 \cdot (-p'_A(0) - p'_B(0)) = 1.$$

Hence, the limit of Eq. (9) is

$$\frac{num'(0)}{den'(0)} = \frac{\beta}{4(\sigma(\beta) - 1/2)} \cdot \left(1 - \gamma + \frac{\sigma(\beta\gamma) - 1/2}{\sigma(\beta) - 1/2}\right) = \frac{\beta}{2} \frac{1 + e^{-\beta}}{1 - e^{-\beta}} \cdot \left(1 - \gamma + \frac{\sigma(\beta\gamma) - 1/2}{\sigma(\beta) - 1/2}\right), \tag{10}$$

which means that each $0 < \gamma < 1$ yields a distortion lower bound for Borda that is larger by a factor of $1 - \gamma + \frac{\sigma(\beta\gamma) - 1/2}{\sigma(\beta) - 1/2}$ than our algorithm-independent lower bound/the upper bound achieved by NLHF. Since this factor is strictly concave in $\gamma$ and is equal to 1 for $\gamma \to 0$ and $\gamma \to 1$, any value of $\gamma$ will lead to a strictly higher bound.

The value of $\gamma$ that maximizes the bound in Eq. (10) is $\gamma^* := \frac{2}{\beta} \operatorname{arctanh}\left(\sqrt{1 - 4\frac{\sigma(\beta) - 1/2}{\beta}}\right)$, which we used to plot Appendix E.2. Since the resulting expression is algebraically unwieldy, we consider the weaker bound for $\gamma = \frac{\log(\beta+1)}{\beta}$, which yields

$$\frac{\beta}{2} \underbrace{\frac{1 + e^{-\beta}}{1 - e^{-\beta}}}_{\to 1 \text{ as } \beta \to \infty} \cdot \underbrace{\left(1 - \frac{\log(\beta + 1)}{\beta} + \frac{\frac{1}{1 + 1/(\beta+1)} - 1/2}{\sigma(\beta) - 1/2}\right)}_{\to 2 \text{ as } \beta \to \infty} = (1 - o(1))\,\beta. \qquad \square$$

### E.3 Algorithm-Independent Lower Bounds

**Theorem 3** (Voting Rule-Independent Distortion Lower Bound). *Fix any $\beta > 0$. If each user provides $d=1$ comparison, no voting rule can guarantee distortion better than $\frac{\beta}{2} \cdot \frac{1 + e^{-\beta}}{1 - e^{-\beta}}$ for large $m$. If each user reports $d \geq 2$ pairwise comparisons, any voting rule that puts at most $1/m$ probability mass on a Condorcet loser must have at least the above distortion.*

*Proof of Theorem 3.* To prove this distortion lower bound, we identify a family of social choice problems for which the distortion of any such social choice function converges towards the claimed bound. We will parameterize these instances by the parameters $m \geq 2$, $0 < \epsilon \leq 1/2$, and $1 \leq \xi < 2$. The instance has $m$ alternatives labeled $a, b_1, \ldots, b_{m-1}$. The distribution $\mathcal{D}$ is such that an agent $i \sim \mathcal{D}$ has utilities

$$(u_i(a), u_i(b_1), \ldots, u_i(b_{m-1})) = \begin{cases} (1, 0, \ldots, 0) & \text{with probability } \frac{\sigma(\beta\epsilon) - 1/2}{\sigma(\beta) + \sigma(\beta\epsilon) - 1} \\ (0, \xi\epsilon, \ldots, \xi\epsilon) & \text{with probability } \frac{\sigma(\beta) - 1/2}{\sigma(\beta) + \sigma(\beta\epsilon) - 1}. \end{cases}$$

Since the $b_j$ alternatives have the same utility for any agent, any agent asked to compare two of them will prefer either one with probability $1/2$. When $\xi = 1$, a randomly drawn rater will prefer alternative $a$ over some alternative $b_j$ with probability

$$\frac{\sigma(\beta\epsilon) - 1/2}{\sigma(\beta) + \sigma(\beta\epsilon) - 1} \cdot \sigma(\beta) + \frac{\sigma(\beta) - 1/2}{\sigma(\beta) + \sigma(\beta\epsilon) - 1} \cdot \sigma(-\beta\epsilon)$$

$$= \frac{\sigma(\beta\epsilon)\sigma(\beta) - \sigma(\beta)/2 + \sigma(\beta)(1 - \sigma(\beta\epsilon)) - (1 - \sigma(\beta\epsilon))/2}{\sigma(\beta) + \sigma(\beta\epsilon) - 1}$$

$$= \frac{\sigma(\beta)/2 + \sigma(\beta\epsilon)/2 - 1/2}{\sigma(\beta) + \sigma(\beta\epsilon) - 1} = 1/2.$$

It is easy to see that the probability of a random agent preferring $a$ over $b_j$ is monotone decreasing in $\xi$. The social welfare of $a$ is clearly $\frac{\sigma(\beta\epsilon)-1/2}{\sigma(\beta)+\sigma(\beta\epsilon)-1}$ and the social welfare of any $b_j$ is $\xi\epsilon\frac{\sigma(\beta)-1/2}{\sigma(\beta)+\sigma(\beta\epsilon)-1}$.

Fix a voting rule $f$. If each agent only provides a single pairwise comparison, the voting rule simply observes $n$ independent Bernoulli samples with bias $1/2$. For any number of samples $n$, denote by $p_x$ the probability that alternative $x$ will win, where the randomness is taken over the realization of these samples and the randomness in $f$. By the pigeon-hole principle, some alternative $x$ must be chosen with probability at most $1/m$ for infinitely many $n$. Without loss of generality, we can assume that this alternative is $a$ (otherwise, simply permute the roles of the alternatives, which does not change the distribution over observed samples), and we restrict our focus to just the $n$ where $p_a \leq 1/m$. Now, the expected social welfare achieved by $f$ is at most

$$\frac{1}{m}\mathsf{AvgUtil}(a) + \mathsf{AvgUtil}(b_1) = \frac{1/m \cdot (\sigma(\beta\epsilon) - 1/2) + \xi\epsilon \cdot (\sigma(\beta) - 1/2)}{\sigma(\beta) + \sigma(\beta\epsilon) - 1}.$$

This shows that the distortion is at least

$$\frac{\mathsf{AvgUtil}(a)}{\mathsf{AvgUtil}(f)} \geq \frac{\sigma(\beta\epsilon) - 1/2}{1/m \cdot (\sigma(\beta\epsilon) - 1/2) + \xi\epsilon \cdot (\sigma(\beta) - 1/2)} = \left(\frac{1}{m} + \frac{\xi\epsilon(\sigma(\beta)-1/2)}{\sigma(\beta\epsilon)-1/2}\right)^{-1}. \tag{11}$$

For a sequence of social choice problems in which $m \to \infty$, $\epsilon \to 0$, and $\xi = 1$, this term converges towards

$$\left(0 + (\sigma(\beta) - 1/2) \cdot \lim_{\epsilon\to 0}\frac{\epsilon}{\sigma(\beta\epsilon)-1/2}\right)^{-1} = \left((\sigma(\beta) - 1/2) \cdot \frac{4}{\beta}\right)^{-1} = \frac{\beta}{2}\frac{1+e^{-\beta}}{1-e^{-\beta}},$$

where the first equality follows from l'Hôpital's rule and the Taylor approximation $\sigma(t) = 1/2 + t/4 + O(t^3)$, and the second inequality follows from the identity $\sigma(t) - 1/2 = \frac{1}{1+e^{-t}} - 1/2 = \frac{2-1-e^{-t}}{2(1+e^{-t})} = \frac{1}{2} \cdot \frac{1-e^{-t}}{1+e^{-t}}$. This shows the claimed bound on the distortion of any voting rule.

If each agent may provide several pairwise comparisons, the above argument does not work for all voting rules. The reason is that the correlations inside an agent's comparisons might lead to nonzero covariances that might allow an (arguably unnatural) voting rule to distinguish the special alternative $a$. If the voting rule satisfies some natural social-choice properties, however, the lower bound above goes through by just slightly changing $\xi$ away from 1.

Suppose, first, that the voting rule satisfies the probabilistic Condorcet loser criterion, i.e., it will never put more than $1/m$ probability mass on a Condorcet loser if one exists. If $\xi > 1$, a random agent prefers $a$ over $b_j$ with less than $1/2$ probability. As a result, as the number of samples grows large, the probability that $a$ is a Condorcet loser with probability converging to 1. Hence, $f$ cannot put more than $1/m$ probability mass on $a$, and the distortion lower bound in Eq. (11) holds. If $\xi$ approaches 1 from above as $m \to \infty$ and $\epsilon \to 0$, the distortion bounds converge to the same limit. $\square$

In the lower bound above, the probabilistic Condorcet loser criterion can easily be replaced by other axioms. If, for example, the voting rule is guaranteed to put at least $1 - 1/m$ probability mass on a Condorcet winner (if one exists), the proof goes through if we increase only $b_2$'s utility by a factor $\xi \searrow 1$.

## F   Supplemental Materials for Section 4

### F.1   Upper Bound for NLHF

**Theorem 7** (NLHF Distortion Upper Bound). *For any instance $\mathcal{D}$ and any $m$, data distribution $\mu$, temperature $\beta$ of the Bradley-Terry model, and any reference policy $\pi_{ref}$ and KL budget $\tau$, we have $dist(\mathsf{NLHF}) \leq \frac{\beta}{2} \cdot \frac{1+e^{-\beta}}{1-e^{-\beta}}$. In the finite-sample regime, we have*

$$\mathsf{AvgUtil}_n(\mathsf{NLHF}) \geq \left(\frac{2}{\beta} \cdot \frac{1-e^{-\beta}}{1+e^{-\beta}}\right) \cdot \max_{\pi^\star \in B_\tau(\pi_{ref})} \mathsf{AvgUtil}(\pi^\star) - O\left(\frac{1}{\beta}\sqrt{\frac{\log(mn)}{n\cdot\min\{1, d\cdot\mu_{\min}^2\}}} + \frac{\log(mn)}{n\cdot\beta\mu_{\min}^2}\right).$$

*Proof of Theorem 7.* We prove this theorem by leveraging the convergence of empirical win-rates in Lemma 10. We first condition on the following successful event, which, according to Lemma 10,

holds with probability at least $1 - \delta$ over $n$ samples of preference data,

$$\forall x, y \in A, \quad |p_n(x \succ y) - p(x \succ y)| \leq O\left(\sqrt{\frac{\log(m/\delta)}{n \cdot \min\{1, d \cdot \mu_{\min}^2\}}} + \frac{\log(m/\delta)}{n\mu_{\min}^2}\right) := \varepsilon_{n,d}(\delta).$$

As argued in the proof sketch, the NLHF policy by definition satisfies

$$\pi_{\mathsf{NLHF}} \in \underset{\pi_1 \in B_\tau(\pi_{\mathrm{ref}})}{\operatorname{argmax}} \min_{\pi_2 \in B_\tau(\pi_{\mathrm{ref}})} \mathbb{E}_{x_1 \sim \pi_1, x_2 \sim \pi_2}[p_n(x_1 \succ x_2) - p_n(x_2 \succ x_1)],$$

where $p_n(x_1 \succ x_2) - p_n(x_2 \succ x_1)$ describes a Nash-equilibrium strategy for a symmetric two-player zero-sum game, and thus have value 0. Therefore, for $\pi^\star \in B_\tau(\pi_{\mathrm{ref}})$, it must hold that

$$0 \leq \underset{x_1 \sim \pi_{\mathsf{NLHF}}, x_2 \sim \pi^\star}{\mathbb{E}}[p_n(x_1 \succ x_2) - p_n(x_2 \succ x_1)] = \underset{x_1 \sim \pi_{\mathsf{NLHF}}, x_2 \sim \pi^\star}{\mathbb{E}}[2p_n(x_1 \succ x_2) - 1]$$

Since $|p_n(x \succ y) - p(x \succ y)| \leq \varepsilon_{n,d}(\delta)$, we have

$$\leq \underset{x_1 \sim \pi_{\mathsf{NLHF}}, x_2 \sim \pi^\star}{\mathbb{E}}[2p(x_1 \succ x_2) - 1] + 2\varepsilon_{n,d}(\delta)$$

According to the linearization lemma (Lemma 1), we have

$$\leq \underset{x_1 \sim \pi_{\mathsf{NLHF}}, x_2 \sim \pi^\star}{\mathbb{E}}[2\beta(L \cdot \mathsf{AvgUtil}(x_1) - \ell_\beta \cdot \mathsf{AvgUtil}(x_2))] + 2\varepsilon_{n,d}(\delta)$$

$$\leq 2\beta(L \cdot \mathsf{AvgUtil}(\pi_{\mathsf{NLHF}}) - \ell_\beta \cdot \mathsf{AvgUtil}(\pi^\star) + \frac{\varepsilon_{n,d}(\delta)}{\beta}).$$

Therefore, under the successful event, we can lower bound the welfare of the NLHF policy by

$$\mathsf{AvgUtil}(\pi_{\mathsf{NLHF}}) \geq \frac{\ell_\beta}{L}\mathsf{AvgUtil}(\pi^\star) - \frac{4\varepsilon_{n,d}(\delta)}{\beta}.$$

Taking the failure event into account, the expected welfare of the NLHF method is at least

$$\mathsf{AvgUtil}_n(\mathsf{NLHF}) \geq (1 - \delta)\left(\frac{\ell_\beta}{L}\mathsf{AvgUtil}(\pi^\star) - \frac{4\varepsilon_{n,d}(\delta)}{\beta}\right)$$

$$\geq \frac{\ell_\beta}{L} \cdot \mathsf{AvgUtil}(\pi^\star) - O\left(\frac{\varepsilon_{n,d}(\delta)}{\beta} + \delta \cdot \frac{\ell_\beta}{L}\right)$$

Finally, choosing $\delta = \Theta\left(\frac{1}{\sqrt{n}}\right)$, we have

$$\geq \frac{\ell_\beta}{L} \cdot \mathsf{AvgUtil}(\pi^\star) - O\left(\frac{1}{\beta}\sqrt{\frac{\log(mn)}{n \cdot \min\{1, d \cdot \mu_{\min}^2\}}} + \frac{\log(mn)}{n \cdot \beta\mu_{\min}^2}\right).$$

This completes the proof. $\qquad\square$

### F.2  Lower Bound for PPO-based RLHF and DPO

**Theorem 6** (RLHF Distortion Lower Bound). *For $m \geq 3$, there is a sequence of alignment problems on which the distortion of RLHF scales as $e^{\Omega(\beta)}$ in $\beta$.*

*Proof of Theorem 6.* Suppose that the instance has $m$ alternatives $A = \{a, b, c_1, \ldots, c_{m-2}\}$, where $m - 2 \geq 4e^\beta$. Let the data collection distribution be uniform over all candidates, i.e., $\mu = \mathsf{Uniform}(A)$. We consider the following distribution $\mathcal{D}$ over utility vectors, such that the utility vector of a random agent $i \sim \mathcal{D}$ satisfies

$$(u_i(a), u_i(b), u_i(c_1), \ldots, u_i(c_{m-2})) = \begin{cases} (0, 1, 0, \ldots, 0) & \text{(type I) with probability } \delta, \\ \left(\frac{1}{\beta}, 0, 1, \ldots, 1\right) & \text{(type II) with probability } 1 - \delta, \end{cases}$$

where $\delta = \frac{10}{10 + e^\beta} = \Theta(e^{-\beta})$. In other words, type I users have a strong preference for candidate $b$ but only constitute a $\delta$ fraction of the population, while type II users have a strong preference for $c$ and weak preference for $a \succ b$ and make up for a $1 - \delta$ fraction of the population.

For the reference policy and the KL budget, we set $\pi_{\mathrm{ref}}(a) = \pi_{\mathrm{ref}}(b) = \frac{1 - \varepsilon}{2}$ and $\pi_{\mathrm{ref}}(c_i) = \frac{\varepsilon}{m-2}$ for all $i \in [m-2]$. We leave the choice of $\varepsilon$ to be determined later. The KL budget $\tau$ is set to be $\tau = 1$.

**Analysis of the MLE reward.** Now we show that when $n \to \infty$, the MLE reward satisfies $r(b) - r(a) > 0$. According to [46, 42], it suffices to show that $\lim_{n\to\infty} \mathsf{BC}_n(b) - \mathsf{BC}_n(a) > 0$, which, by Lemma 11, is implied by $\mathsf{BC}^\star(b) - \mathsf{BC}^\star(a) > 0$.

We have

$$\mathsf{BC}^\star(b) - \mathsf{BC}^\star(a) = \frac{1}{m} \sum_{i=1}^{m-2} (p(b \succ c_i) - p(a \succ c_i)) + \frac{1}{m} (p(b \succ a) - p(a \succ b))$$

For each $c_i$, we have

$$p(b \succ c_i) - p(a \succ c_i) = \delta \cdot (\sigma(\beta) - \sigma(0)) + (1 - \delta) \cdot (\sigma(-\beta) - \sigma(1 - \beta)).$$

In the above equation, the first term accounts for type-I users and is lower bounded by $\frac{\delta}{3}$ when $\beta \geq 2$. The second term accounts for type-II users, and we leverage the fact that $\sigma(x)$ is concave when $x \geq 0$ to bound it as

$$(1 - \delta) \cdot (\sigma(-\beta) - \sigma(1 - \beta)) = -\frac{e^\beta}{10} \delta \cdot (\sigma(\beta) - \sigma(\beta - 1)) \geq -\frac{e^\beta}{10} \delta \cdot \sigma'(\beta - 1) \geq -\frac{e}{10} \delta,$$

where the last step uses $\sigma'(x) = \sigma(x) \cdot (1 - \sigma(x)) \leq 1 - \sigma(x) \leq e^{-x}$ for all $x$. Plugging both bounds into the limit $\mathsf{BC}^\star(b) - \mathsf{BC}^\star(a)$ and substituting $\delta = \frac{10}{10+e^\beta} \geq \frac{20}{m-2}$ gives

$$\lim_{n\to\infty} \mathsf{BC}(b) - \mathsf{BC}(a) = \mathsf{BC}^\star(b) - \mathsf{BC}^\star(a) \geq \frac{m-2}{m} \cdot \delta \cdot \left(\frac{1}{3} - \frac{e}{10}\right) - \frac{1}{m} \geq \frac{1}{m}.$$

Therefore, when $n$ is sufficiently large, we have $\mathsf{BC}(b) > \mathsf{BC}(a)$ with high probability, which implies that $b$ is has higher MLE reward than $a$.

As for the MLE reward of type-$c$ candidates, since $\mathsf{BC}^\star(c_i) = \mathsf{BC}^\star(c_j)$ for all $i, j \in [m-2]$, we have $\max_{i,j\in[m-2]} |r(c_i) - r(c_j)| \to 0$ when $n \to \infty$. In the limit, we can treat all type-$c$ candidates as having the same reward.

**Analysis of the KL-constrained policies.** Since all type-$c$ candidates have the same estimated reward and the same probability under the reference policy, both $\pi^\star$ and $\hat{\pi}_{\mathsf{RLHF}}$ will assign the same probability to all type-$c$ candidates. This can be seen by the equivalence between regularized and constrained RLHF as shown in Appendix F.4. As a result, we can view all type-$c$ candidates as a single candidate $c$ which have mass $\varepsilon$ under the reference policy.

We now show that for any $\eta > 0$, there exists an $\varepsilon > 0$ such that any policy $\pi \in \Delta(\{a, b, c\})$ inside the KL ball $B_\tau(\pi_{\mathrm{ref}})$ cannot put more than $\eta$ mass on $c$. We have

$$1 \geq D_{\mathrm{KL}}(\pi \| \pi_{\mathrm{ref}}) = \pi(a) \cdot \log \frac{\pi(a)}{(1-\varepsilon)/2} + \pi(b) \cdot \log \frac{\pi(b)}{(1-\varepsilon)/2} + \pi(c) \cdot \log \frac{\pi(c)}{\varepsilon}$$

Fixing $\pi(c)$, the KL divergence is minimized when $\pi(a) = \pi(b) = \frac{1-\pi(c)}{2}$. Substituting this into the KL divergence, we get

$$\geq (1 - \pi(c)) \cdot \log \frac{1 - \pi(c)}{1 - \varepsilon} + \pi(c) \cdot \log \frac{\pi(c)}{\varepsilon}$$

Since $t \log t \geq -1/e$ for all $t > 0$, and $\log \frac{1}{1-\varepsilon} > 0$, we have

$$\geq -\frac{2}{e} + \pi(c) \log \frac{1}{\varepsilon}.$$

Therefore, any policy in the KL ball must satisfy $\pi(c) \log \frac{1}{\varepsilon} \leq 1 + 2/e \leq 2$, which implies that $\pi(c) \leq \frac{2}{\log(1/\varepsilon)}$. We can choose $\varepsilon$ to be any constant smaller than $e^{-2/\eta}$ to ensure that $\pi(c) \leq \eta$.

We then show that when $\eta$ is sufficiently small, $\pi_{\mathsf{RLHF}}$ puts almost all probability mass on $b$, whereas $\pi^\star$ puts almost all probability mass on $a$. This will ultimately lead to a distortion of

$$\frac{\mathsf{AvgUtil}(\pi^\star)}{\mathsf{AvgUtil}(\hat{\pi}_{\mathsf{RLHF}})} = \frac{\Theta(\mathsf{AvgUtil}(a))}{\Theta(\mathsf{AvgUtil}(b))} = \frac{\Theta(1/\beta)}{\Theta(e^{-\beta})} = e^{\Omega(\beta)}.$$

- For $\pi_{\mathsf{RLHF}}$, we assume that the estimated reward is shifted such that $r(c) = 0$ (as a result, $r(a) < r(b) < 0$). Since $\pi' = (0, 1, 0)$ also satisfies the KL constraint, we have $r(\pi_{\mathsf{RLHF}}) \geq r(\pi')$. Together with the fact that $\pi_{\mathsf{RLHF}}(c) \leq \eta$, we have

$$r(\pi') = r(b) \leq r(\pi_{\mathsf{RLHF}}) = r(a)\pi_{\mathsf{RLHF}}(a) + r(b)\pi_{\mathsf{RLHF}}(b)$$
$$\leq \pi_{\mathsf{RLHF}}(a) \cdot r(a) + (1 - \pi_{\mathsf{RLHF}}(a) - \eta) \cdot r(b).$$

Therefore, we have $\pi_{\mathsf{RLHF}}(a) \leq \eta \cdot \frac{|r(b)|}{|r(b) - r(a)|}$. Setting $\varepsilon$ to be sufficiently small, we can guarantee that

$$\eta \leq \eta_1 := \frac{e^{-\beta}}{1 + \frac{|r(b)|}{|r(b) - r(a)|}}, \tag{12}$$

and thus $\pi_{\mathsf{RLHF}}(a) + \pi_{\mathsf{RLHF}}(c) \leq \eta \left(1 + \frac{|r(b)|}{|r(b) - r(a)|}\right) \leq e^{-\beta}$. As a result, we have $\mathsf{AvgUtil}(\pi_{\mathsf{RLHF}}) = \Theta(\mathsf{AvgUtil}(b))$.

- For $\pi^\star$, a similar argument shows that when

$$\eta \leq \eta_2 := \frac{e^{-\beta}}{1 + \frac{\mathsf{AvgUtil}(a)}{\mathsf{AvgUtil}(a) - \mathsf{AvgUtil}(b)}}, \tag{13}$$

We have $\pi^\star(b) + \pi^\star(c) \leq e^{-\beta}$. As a result, we have $\mathsf{AvgUtil}(\pi^\star) = \Theta(\mathsf{AvgUtil}(a))$.

Finally, we set $\varepsilon$ to be smaller than $e^{-2/\min\{\eta_1, \eta_2\}}$ such that Equations (12) and (13) are both satisfied. This ensures $\mathsf{AvgUtil}(\pi^\star)/\mathsf{AvgUtil}(\pi_{\mathsf{RLHF}}) = e^{\Omega(\beta)}$ and completes the proof. $\qquad\square$

### F.3 Equivalence of DPO and RLHF under Heterogeneous Preferences

In this section, we formalize the observation that DPO and RLHF are equivalent under heterogeneous preferences. This is consistent with the result by Shirali et al. [45], which shows that DPO also aligns with the Borda count. We start by recalling the DPO objective [43].[12]

$$\mathcal{L}_{\mathsf{DPO}}(\pi; \pi_{\mathsf{ref}}) = -\sum_{1 \leq i \leq n, 1 \leq j \leq d} \log \sigma \left(\beta \log \frac{\pi(x_i^j)}{\pi_{\mathsf{ref}}(x_i^j)} - \beta \log \frac{\pi(y_i^j)}{\pi_{\mathsf{ref}}(y_i^j)}\right).$$

Now we perform a change of variables to transform $\pi$ into the following form:

$$\pi(x) = \pi_{\mathsf{ref}}(x) \cdot \exp(\hat{r}(x)/\beta) \quad \text{where} \quad \hat{r}(x) := \beta \log \frac{\pi(x)}{\pi_{\mathsf{ref}}(x)}, \ \forall x \in A.$$

Substituting this into the DPO objective, we get:

$$\mathcal{L}_{\mathsf{DPO}}(\pi; \pi_{\mathsf{ref}}) = -\sum_{1 \leq i \leq n, 1 \leq j \leq d} \log \left(\sigma(\hat{r}(x_i^j) - \hat{r}(y_i^j))\right),$$

which is exactly the MLE objective for reward learning in RLHF, repeated here for convenience:

$$\mathcal{L}_{\mathsf{MLE}}(r) := -\sum_{1 \leq i \leq n, 1 \leq j \leq d} \log \left(\sigma(r(x_i^j) - r(y_i^j))\right).$$

Therefore, there is a one-to-one correspondence between the MLE reward $r^\star = \arg\min_{r \in \mathbb{R}^m} \mathcal{L}_{\mathsf{MLE}}(r)$[13], and the DPO policy $\pi_{\mathsf{DPO}} = \arg\min_\pi \mathcal{L}_{\mathsf{DPO}}(\pi; \pi_{\mathsf{ref}})$ as:

$$\pi_{\mathsf{DPO}}(x) = \pi_{\mathsf{ref}}(x) \cdot \exp(r^\star(x)/\beta).$$

On the other hand, the RLHF policy with regularization parameter $\lambda$ (see Appendix F.4 for the equivalence between regularized and constrained versions of RLHF) is also given by

$$\pi_{\mathsf{RLHF}}(x) = \pi_{\mathsf{ref}}(x) \cdot \exp(r^\star(x)/\lambda).$$

Therefore, the DPO policy $\pi_{\mathsf{DPO}}$ and the RLHF policy $\pi_{\mathsf{RLHF}}$ are equivalent when the parameter $\beta$ in the DPO objective is equal to $\lambda$ in the RLHF objective. Notably, both policies are different from the optimal policy $\pi^\star \propto \pi_{\mathsf{ref}} \cdot \exp(\mathsf{AvgUtil}(x)/\lambda')$ (for a potentially different $\lambda'$ that make the KL constraint tight) as $r^\star \neq \mathsf{AvgUtil}$, which has also been pointed out by Shirali et al. [45].

---

[12]Note that the parameter $\beta$ does not need to be the same as the true temperature of the Bradley-Terry model in our setting.

[13]Note that we have a minus sign in the MLE objective, which is equivalent to maximizing the sum of log-likelihoods.

### F.4 Equivalence of Regularized and Constrained Alignment Methods

In this section, we formally establish the equivalence between regularized and constrained formulations of both RLHF and NLHF. Although this equivalence is standard and likely known to many, we include the details here for completeness. We begin by proving the equivalence between the two versions of NLHF (which involves max-min optimization over the policy space); the corresponding result for RLHF then follows by analogous arguments.

**Proposition 13** (Equivalence between Constrained and Regularized NLHF.). *Let $\pi_\tau$ be the output of the $\tau$-constrained NLHF method defined in Section 2, i.e.,*

$$\pi_\tau = \underset{\pi_1 \in B_\tau(\pi_{ref})}{\operatorname{argmax}} \min_{\pi_2 \in B_\tau(\pi_{ref})} \underset{x_1 \sim \pi_1, x_2 \sim \pi_2}{\mathbb{E}} [M_{x_1, x_2}], \tag{14}$$

*and let $\tilde{\pi}_\lambda$ be the output of the $\lambda$-regularized NLHF method [36], i.e.,*

$$\tilde{\pi}_\lambda = \underset{\pi_1 \in \Delta(M)}{\operatorname{argmax}} \min_{\pi_2 \in \Delta(M)} \underset{x_1 \sim \pi_1, x_2 \sim \pi_2}{\mathbb{E}} [M_{x_1, x_2}] - \lambda \cdot D_{KL}(\pi_1 \,\|\, \pi_{ref}) + \lambda \cdot D_{KL}(\pi_2 \,\|\, \pi_{ref}). \tag{15}$$

*Then, for each $\lambda \in [0, \infty]$ with solution $\tilde{\pi}_\lambda$ for Eq. (15), we have that $\tilde{\pi}_\lambda$ is also an optimal solution to the $\tau$-constrained optimization problem in Eq. (14), where $\tau = D_{KL}(\tilde{\pi}_\lambda \,\|\, \pi_{ref})$. Conversely, for each $\tau \geq 0$ with solution $\pi_\tau$ for Eq. (14), there exists $\lambda \in [0, \infty]$ such that $\pi_\tau$ is also an optimal solution to the $\lambda$-regularized optimization problem in Eq. (15).*

*Proof of Proposition 13.* We start by observing that in both games Eq. (14) and Eq. (15), the utilities are anti-symmetric functions, and the strategy spaces for both players are identical, convex and compact. Therefore, the value of the both games is 0, and both games have symmetric Nash equilibria.

Now, we prove the two directions of the claim separately.

**Regularized $\Rightarrow$ Constrained.** Given $\lambda \in [0, \infty]$ and $\tilde{\pi}_\lambda$ be the solution to Eq. (15). Consider the constrained optimization problem in Eq. (14) with $\tau = D_{KL}(\tilde{\pi}_\lambda \,\|\, \pi_{ref})$. We show that $\pi_2 = \tilde{\pi}_\lambda$ is a best response to $\pi_1 = \tilde{\pi}_\lambda$ in the constrained game with radius $\tau = D_{KL}(\tilde{\pi}_\lambda \,\|\, \pi_{ref})$, i.e.,

$$\tilde{\pi}_\lambda = \underset{\pi_2 \in B_\tau(\pi_{ref})}{\operatorname{argmin}} \underset{x_1 \sim \tilde{\pi}_\lambda, x_2 \sim \pi_2}{\mathbb{E}} [M_{x_1, x_2}] \tag{16}$$

If Eq. (16) holds, then by the fact that the utility is anti-symmetric, we have that $\pi_1 = \tilde{\pi}_\lambda$ is also a best response to $\pi_2 = \tilde{\pi}_\lambda$ in the same constrained game. Putting both together, we have that $(\tilde{\pi}_\lambda, \tilde{\pi}_\lambda)$ is a Nash equilibrium for the constrained game with radius $\tau = D_{KL}(\tilde{\pi}_\lambda \,\|\, \pi_{ref})$, thus establishing the first direction.

Now we prove Equation (16). To see this, note that $\forall \pi_2 \in B_\tau(\pi_{ref})$, we have that $D_{KL}(\pi_2 \,\|\, \pi_{ref}) \leq \tau = D_{KL}(\tilde{\pi}_\lambda \,\|\, \pi_{ref})$. Therefore, from the fact that $\tilde{\pi}_\lambda$ is a Nash equilibrium for the regularized game Eq. (15), we have that for any $\pi_2 \in \Delta(M)$ and specifically $\pi_2 \in B_\tau(\pi_{ref})$, we have that

$$\underset{x_1 \sim \tilde{\pi}_\lambda, x_2 \sim \pi_2}{\mathbb{E}} [M_{x_1, x_2}] + \lambda \cdot D_{KL}(\pi_2 \,\|\, \pi_{ref}) \geq \underset{x_1 \sim \tilde{\pi}_\lambda, x_2 \sim \tilde{\pi}_\lambda}{\mathbb{E}} [M_{x_1, x_2}] + \lambda \cdot D_{KL}(\tilde{\pi}_\lambda \,\|\, \pi_{ref})$$

$$\Rightarrow \underset{x_1 \sim \tilde{\pi}_\lambda, x_2 \sim \pi_2}{\mathbb{E}} [M_{x_1, x_2}] - \underset{x_1 \sim \tilde{\pi}_\lambda, x_2 \sim \tilde{\pi}_\lambda}{\mathbb{E}} [M_{x_1, x_2}] \geq \lambda \cdot (D_{KL}(\tilde{\pi}_\lambda \,\|\, \pi_{ref}) - D_{KL}(\pi_2 \,\|\, \pi_{ref})) \geq 0.$$

This proves Eq. (16) and completes the proof of the first direction.

**Constrained $\Rightarrow$ Regularized.** We prove the reverse direction using duality theory.

We first show how to construct the regularization parameter $\lambda$. For simplicity, we write $\pi_1^\mathsf{T} M \pi_2$ as a shorthand for $\mathbb{E}_{x_1 \sim \pi_1, x_2 \sim \pi_2}[M_{x_1, x_2}]$. Then minimizing player in the constrained game Eq. (14) with $\pi_1 = \pi_\tau$ can be written as

$$\min_{\pi_2 \in \mathbb{R}^m} \pi_\tau^\mathsf{T} M \pi_2 \quad \text{s.t.} \quad D_{KL}(\pi_2 \| \pi_{ref}) \leq \tau, \pi_2 \geq \mathbf{0}, \mathbf{1}^\mathsf{T} \pi_2 = 1. \qquad \text{(Constrained Minimization)}$$

The Lagrangian of this problem is

$$\mathcal{L}(\pi_2, \lambda, \vec{\mu}, \eta) = \pi_\tau^\mathsf{T} M \pi_2 + \lambda (D_{KL}(\pi_2 \| \pi_{ref}) - \tau) - \vec{\mu}^\mathsf{T} \pi_2 + \eta \mathbf{1}^\mathsf{T} \pi_2,$$

where $\lambda, \vec{\mu}, \eta \geq 0$ are the Lagrange multipliers for the KL divergence constraint, the non-negativity constraint, and the normalization constraint, respectively. Since the utility $\pi_1^\mathsf{T} M \pi_2$ is convex, the normalization constraint is affine, and the inequality constraints are convex, and the reference policy $\pi_{\text{ref}}$ is strictly feasible with $D_{\mathsf{KL}}(\pi_{\text{ref}} \| \pi_{\text{ref}}) = 0 < \tau$,[14] the Slater's condition is satisfied, which guarantees that the KKT conditions are necessary and sufficient for optimality — there exists parameters $\lambda^\star, \vec{\mu}^\star, \eta^\star \geq 0$ such that as an optimal solution to Eq. (Constrained Minimization), $\pi_2 = \pi_\lambda$ satisfies the following KKT conditions:

$$\nabla_{\pi_2} \mathcal{L}(\pi_2, \lambda^\star, \vec{\mu}^\star, \eta^\star) = M^\mathsf{T} \pi_\tau + \lambda^\star \cdot \nabla_{\pi_2} D_{\mathsf{KL}}(\pi_2 \| \pi_{\text{ref}}) - \vec{\mu}^\star + \eta^\star \cdot \mathbf{1} = 0, \qquad (17)$$

where

$$\eta^\star(\mathbf{1}^\mathsf{T} \pi_2 - 1) = 0 \quad \text{and} \quad \mu_i^\star(\pi_2)_i = 0, \forall i \in [m], \qquad (18)$$

due to complementary slackness.

We will show that this $\lambda^\star$ is the regularization parameter we are looking for. Namely, for the regularized game in Eq. (15) with $\lambda = \lambda^\star$, we have that $(\pi_\tau, \pi_\tau)$ is a Nash equilibrium.

Again, let us first fix $\pi_1 = \pi_\tau$ and consider the regularized optimization problem for the minimizing player:

$$\min_{\pi_2} \pi_\tau^\mathsf{T} M \pi_2 - \lambda^\star \cdot D_{\mathsf{KL}}(\pi_2 \| \pi_{\text{ref}}) \quad \text{s.t.} \quad \pi_2 \geq \mathbf{0}, \mathbf{1}^\mathsf{T} \pi_2 = 1. \qquad \text{(Regularized Minimization)}$$

Since $\pi_\tau^\mathsf{T} M \pi_2$ is convex in $\pi_2$, Slater's condition is satisfied for the above problem and implies that the KKT conditions are sufficient for optimality. It is also not hard to see that $\mathcal{L}(\pi_2, \lambda^\star, \vec{\mu}^\star, \eta^\star)$ coincides with the Lagrangian of (Regularized Minimization). We can therefore conclude from Eqs. (17) and (18) that for the minimizing player in the regularized game, $\pi_2 = \pi_\tau$ is a best response strategy to $\pi_1 = \pi_\tau$.

Since the regularized game is anti-symmetric, the same argument shows that for the maximizing player in the regularized game, $\pi_1 = \pi_\tau$ is also a best response strategy to $\pi_2 = \pi_\tau$. Together, we have that $(\pi_\tau, \pi_\tau)$ is a Nash equilibrium for the regularized game in Eq. (15) with $\lambda = \lambda^\star$. The proof is complete. $\qquad \square$

Combining Proposition 13 with Theorem 7, we obtain the following guarantee for the KL-regularized version of NLHF:

**Corollary 8.** *If $\lambda$-regularized NLHF (for any $\lambda \geq 0$) returns a policy $\tilde{\pi}_{\mathsf{NLHF}}$, this policy's average utility is at least a $\frac{2}{\beta} \cdot \frac{1 - e^{-\beta}}{1 + e^{-\beta}}$ fraction of the optimal average utility of any policy $\pi$ with $D_{KL}(\pi \| \pi_{ref}) \leq D_{KL}(\tilde{\pi}_{\mathsf{NLHF}} \| \pi_{ref})$ (minus finite-sample errors, see Theorem 7).*

For the RLHF case, the proof is analogous, except that we no longer have nested minimization-maximization, so the proof is slightly simpler. We omit the details of proof, but state the result below.

**Proposition 14** (Equivalence between Constrained and Regularized RLHF)**.** *Let $r$ be the MLE reward learned from the comparison data, and let $\pi_\tau$ be the output of the $\tau$-constrained RLHF method defined in Section 2, i.e.,*

$$\pi_\tau = \underset{\pi \in B_\tau(\pi_{ref})}{\arg\max} \underset{x \sim \pi}{\mathbb{E}} [r(x)], \qquad (19)$$

*and let $\tilde{\pi}_\lambda$ be the output of the $\lambda$-regularized RLHF method, i.e.,*

$$\tilde{\pi}_\lambda = \underset{\pi \in \Delta(M)}{\arg\max} \underset{x \sim \pi}{\mathbb{E}} [r(x)] - \lambda \cdot D_{KL}(\pi \| \pi_{ref}). \qquad (20)$$

*Then, for each $\lambda \in [0, \infty]$ with solution $\tilde{\pi}_\lambda$ for Eq. (20), we have that $\tilde{\pi}_\lambda$ is also an optimal solution to the $\tau$-constrained optimization problem in Eq. (19), where $\tau = D_{KL}(\tilde{\pi}_\lambda \| \pi_{ref})$. Conversely, for each $\tau \geq 0$ with solution $\pi_\tau$ for Eq. (19), there exists $\lambda \in [0, \infty]$ such that $\pi_\tau$ is also an optimal solution to the $\lambda$-regularized optimization problem in Eq. (20).*

---

[14]If $\tau = 0$, the only feasible policy is $\pi_{\text{ref}}$, and the claim clearly holds for $\lambda = \infty$. We also assume that $\pi_{\text{ref}}(a) > 0$ for all $a \in M$. Otherwise if $\pi_{\text{ref}}(a) = 0$ for some $a \in M$, then both the regularized and constrained versions forbid any policy to put nonzero probability on $a$, which leads to an effectively smaller candidate set.

# G Other Sampling Models

To prove Theorem 9, we first prove the following lemma. We illustrate the constructed sequence of alternatives in Fig. 4.

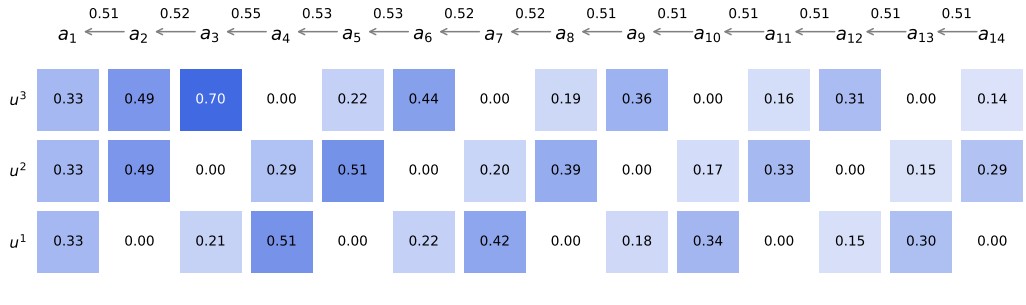

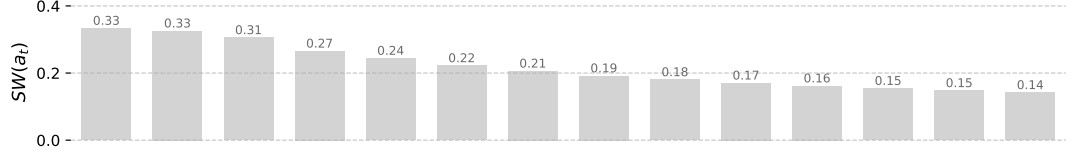

Figure 4: Utilities for first 14 alternatives in the sequences constructed in Lemma 15, for $\beta = 5$. Bottom bar chart shows decreasing welfare. Numbers between alternative labels $a_{t+1} \to a_t$ give the expected win-rate $p(a_{t+1} \succ a_t)$.

**Lemma 15.** *For any $\beta > 0$, there is an infinite sequence $a_1, a_2, \ldots$ of alternatives, and a distribution $\mathcal{D}$ of utility functions over these alternatives such that*

- $\mathsf{AvgUtil}(a_1) = 1/3$,

- *for all $t \geq 2$, $0 < \mathsf{AvgUtil}(a_t) \leq \mathsf{AvgUtil}(a_{t-1}) - \frac{2}{3\beta} \log \left(1 + \tanh \left(\beta/4 \cdot \mathsf{AvgUtil}(a_{t-1})\right)^3\right) < \mathsf{AvgUtil}(a_{t-1})$, and*

- *for all $t \geq 2$, $p(a_t \succ a_{t-1}) > 1/2$.*

*Proof.* Our population $\mathcal{D}$ will be a uniform distribution over three utility vectors, $u^1$, $u^2$, and $u^3$. We define the sequence of alternatives and prove the claim by induction over $t \geq 1$.

For $t = 1$, set $u^1(a_1) = u^2(a_1) = u^3(a_1) \coloneqq 1/3$, which clearly satisfies the first claim.

Now, let $t \geq 2$, and suppose that we have defined the utilities for alternatives $a_1, \ldots, a_{t-1}$ and established the claims for all $t' < t$. We define utilities for $a_t$ and extend the claims to $t$. Let $u^A$ denote the utility vector among $u^1, u^2, u^3$ with the highest utility for $a_{t-1}$, and denote the other two utility vectors by $u^B, u^C$. For convenience, set $\Delta \coloneqq u^A(a_{t-1}) \cdot \beta$ and $\Delta' \coloneqq \log\left(\frac{(e^{\Delta/2}+1)^3}{2\,(e^{\Delta}+3)}\right)$.

$$u^A(a_t) \coloneqq u^A(a_{t-1}) - \Delta/\beta = 0, \qquad u^B(a_t) \coloneqq u^B(a_{t-1}) + \Delta'/\beta, \qquad u^C(a_t) \coloneqq u^C(a_{t-1}) + \Delta'/\beta.$$

It will be useful to derive an alternative expression for $\Delta'$:

$$\begin{aligned}
\Delta' &= \log\left(\frac{(e^{\Delta/2}+1)^3}{2\,(e^{\Delta}+3)}\right) = \log\left(e^{\Delta/2}\,\frac{(e^{\Delta/2}+1)^3}{2\,e^{\Delta/2}\,(e^{\Delta}+3)}\right) \\
&= \frac{\Delta}{2} - \log\frac{2\,e^{\Delta/2}\,(e^{\Delta}+3)}{(e^{\Delta/2}+1)^3} = \frac{\Delta}{2} - \log\frac{(e^{\Delta/2}+1)^3 + (e^{\Delta/2}-1)^3}{(e^{\Delta/2}+1)^3} \\
&= \frac{\Delta}{2} - \log\left(1 + \left(\frac{e^{\Delta/2}-1}{e^{\Delta/2}+1}\right)^3\right) = \frac{\Delta}{2} - \log\left(1 + \tanh(\Delta/4)^3\right).
\end{aligned}$$

Since the value $1 + \left(\frac{e^{\Delta/2}-1}{e^{\Delta/2}+1}\right)^3$ in the logarithm is greater than 1, we know that $\Delta' < \Delta/2$.

Since $\mathsf{AvgUtil}(a_{t-1}) > 0$ by the induction hypothesis, it must hold that $\Delta > 0$, and, by expanding, that

$$\Delta' = \log \frac{(e^{\Delta/2}+1)^3}{2\,(e^\Delta+3)} = \log \frac{3\,e^\Delta+1+\frac{1}{2}\cdot(e^{\Delta/2}-1)^3}{e^\Delta+3} > \log \frac{3\,e^\Delta+1}{e^\Delta+3}. \qquad (21)$$

Since $\log \frac{3\,e^\Delta+1}{e^\Delta+3} > \log \frac{e^\Delta+3}{e^\Delta+3} = 0$, it holds that $\Delta' > 0$ and that $\mathsf{AvgUtil}(a_t) > 0$.

We first must show that we have not set $u^B(a_t)$ and $u^C(a_t)$ greater than 1. Since, by the induction hypothesis, $\frac{1}{3}(u^A(a_{t-1}) + u^B(a_{t-1}) + u^C(a_{t-1})) = \mathsf{AvgUtil}(a_{t-1}) \le \mathsf{AvgUtil}(a_{t-2}) \le \cdots \le \mathsf{AvgUtil}(a_1) = 1/3$, it must hold that $u^A(a_{t-1}) + u^B(a_{t-1}) + u^C(a_{t-1}) \le 1$, which by the choice of $u^A$ implies that $u^B(a_{t-1}), u^C(a_{t-1})$ are at most $1/2$. Since $\Delta' < \Delta/2 = u^A(a_{t-1}) \cdot \beta/2 \le \beta/2$, $u^B(a_t) = u^B(a_{t-1}) + \Delta'/\beta \le 1/2 + 1/2 = 1$, this holds for $u^B$, and analogously for $u^C$.

Next, we show the claimed reduction in social welfare using our alternative expression for $\Delta'$.

$$\mathsf{AvgUtil}(a_t) = \frac{1}{3} \cdot \left(u^A(a_t) + u^B(a_t) + u^C(a_t)\right) = \frac{1}{3} \cdot \left(u^A(a_{t-1}) + u^B(a_{t-1}) + u^C(a_{t-1}) - \frac{\Delta - 2\,\Delta'}{\beta}\right)$$

$$= \mathsf{AvgUtil}(a_{t-1}) - \frac{\Delta - 2\,\Delta'}{3\,\beta} = \mathsf{AvgUtil}(a_{t-1}) - \frac{2}{3\,\beta}\,\log\left(1 + \tanh(\Delta/4)^3\right).$$

By our choice of $u^A$ and averaging, it holds that $u^A(a_{t-1}) \ge \mathsf{AvgUtil}(a_{t-1})$ and hence that $\Delta \ge \beta \cdot \mathsf{AvgUtil}(a_{t-1})$. Since the bound on $\mathsf{AvgUtil}(a_t)$ above is monotone nonincreasing in $\Delta$, we obtain our claim that

$$\mathsf{AvgUtil}(a_t) \le \mathsf{AvgUtil}(a_{t-1}) - \frac{2}{3\,\beta}\,\log\left(1 + \tanh\left(\beta/4 \cdot \mathsf{AvgUtil}(a_{t-1})\right)^3\right).$$

Finally, it remains to show that $p(a_t \succ a_{t-1}) > 1/2$. Since

$$p(a_t \succ a_{t-1}) = \frac{\sigma(-\Delta) + 2\,\sigma(\Delta')}{3} = \frac{1}{2} + \frac{(\sigma(-\Delta) - 1/2) + 2\,(\sigma(\Delta') - 1/2)}{3}$$

$$= \frac{1}{2} + \frac{(1/2 - \sigma(\Delta)) + 2\,(\sigma(\Delta') - 1/2)}{3},$$

it suffices to show that $2\,(\sigma(\Delta') - 1/2) > \sigma(\Delta) - 1/2$. Observing that $\sigma(x) - 1/2 = \frac{1}{2} \cdot \frac{1-e^{-x}}{1+e^{-x}}$ and applying Eq. (21), we bound

$$2\left(\sigma(\Delta') - \tfrac{1}{2}\right) > 2\left(\sigma\left(\log(\tfrac{3^\Delta+1}{e^\Delta+3})\right) - \tfrac{1}{2}\right) = \frac{1 - \frac{e^\Delta+3}{3\,e^\Delta+1}}{1 + \frac{e^\Delta+3}{3\,e^\Delta+1}} = \frac{\frac{2\,e^\Delta-2}{3\,e^\Delta+1}}{\frac{4\,e^\Delta+4}{3\,e^\Delta+1}} = \frac{2}{4} \cdot \frac{e^\Delta - 1}{e^\Delta + 1}$$

$$= \frac{1}{2} \cdot \frac{1 - e^{-\Delta}}{1 + e^{-\Delta}} = \sigma(\Delta) - \tfrac{1}{2},$$

which establishes our claim. $\qquad \square$

**Theorem 9** (Unbounded Distortion of RLHF Under Correlated Sampling). *For any $\beta > 0$, there exists a sequence of alignment instances and distributions $\nu \in \Delta\left(\binom{A}{2}\right)$ over comparison pairs such that RLHF's distortion is unbounded.*

*Proof.* We construct our sequence of instances by taking increasingly long prefixes of the sequence in Lemma 15, i.e., by considering the alternatives $a_1, \ldots, a_m$ for increasing $m$. Rescaling by some constants, we can define the MLE rewards in RLHF as

$$r = \underset{r \in \mathbb{R}^m}{\mathrm{argmax}} \sum_{\{x,y\} \in \binom{A}{2}} \frac{\#(x \succ y)}{n\,d}\,\log\left(\sigma(r(x) - r(y))\right) + \frac{\#(y \succ x)}{n\,d}\,\log\left(\sigma(r(y) - r(x))\right).$$

Following Siththaranjan et al. [46], we apply the first-order optimality conditions to obtain that, for each alternative $x$,

$$\sum_{y \neq x} \frac{\#(x \succ y)}{n\,d} = \sum_{y \neq x} \frac{\#(x \succ y) + \#(y \succ x)}{n\,d} \cdot \sigma(r(x) - r(y)).$$

By the strong law of large numbers, as the number $n$ of samples goes to infinity (regardless of $d$), the sample fraction $\frac{\#(x \succ y)}{n \, d}$ converges almost surely to its expected value $\nu(\{x, y\}) \cdot p(x \succ y)$. Hence, as $n \to \infty$, the rewards (a random variable depending on the random pairwise comparisons) will satisfy that

$$\sum_{y \neq x} \nu(\{x, y\}) \cdot \sigma(r(x) - r(y)) \xrightarrow{\text{a.s.}} \sum_{y \neq x} \nu(\{x, y\}) \cdot p(x \succ y). \tag{22}$$

Consider a distribution $\nu$ over pairs of alternatives that assigns each pair of adjacent alternatives $\{a_t, a_{t+1}\}$ a probability of $\frac{1-\epsilon}{m-1}$ of being drawn for comparison, and all other pairs a probability of $\frac{\epsilon}{\binom{m}{2} - (m-1)}$, where $\epsilon > 0$ is a small value, dependent on the current $m$, to be determined in the following.

Applying Eq. (22) to $x = a_m$, we obtain that $\frac{1-\epsilon}{m-1}\sigma(r(a_m) - r(a_{m-1})) + O(\epsilon)$ converges almost surely to $\frac{1-\epsilon}{m-1} \cdot p(a_m \succ a_{m-1}) + O(\epsilon)$. Since Lemma 15 guarantees that $p(a_m \succ a_{m-1}) > 1/2$, for small enough $\epsilon$, it will hold almost surely that $\sigma(r(a_m) - r(a_{m-1})) > 1/2$, i.e., that $r(a_m) > r(a_{m-1})$.

Next, we apply Eq. (22) to $x = a_{m-1}$, to obtain that $\frac{1-\epsilon}{m-1}(\sigma(r(a_{m-1}) - r(a_m)) + \sigma(r(a_{m-1}) - r(a_{m-2}))) + O(\epsilon)$ converges almost surely to $\frac{1-\epsilon}{m-1} \cdot (p(a_{m-1} \succ a_m) + p(a_{m-1} \succ a_{m-2})) + O(\epsilon)$. Having established above that, for small enough $\epsilon$, we can make $\sigma(r(a_{m-1}) - r(a_m)) = 1 - \sigma(r(a_m) - r(a_{m-1}))$ arbitrarily close to $p(a_{m-1} \succ a_m) = 1 - p(a_m \succ a_{m-1})$, we see that $\sigma(r(a_{m-1}) - r(a_{m-2}))$ must become arbitrarily close to $p(a_{m-1} \succ a_{m-2}) > 1/2$.

Continuing this argument for $x = a_{m-2}, a_{m-3}, \ldots, a_1$, we obtain that, for small enough $\epsilon$, the rewards will be ordered as $r(a_1) < r(a_2) < \cdots < r(a_m)$ almost surely. Set $\epsilon$ (for this specific $m$) so that this is the case. We set the KL constraint large enough that all policies are possible; say, by choosing the reference policy to be uniform and setting $\tau = \log m$.[15] Then, the reward-maximizing policy clearly puts all probability mass on the alternative $a_m$ with maximal reward, obtaining a distortion of $\frac{\text{AvgUtil}(a_1)}{\text{AvgUtil}(a_m)} = \frac{1/3}{\text{AvgUtil}(a_m)}$.

We have now defined a sequence of instances, whose distortion grows as $\frac{1/3}{\text{AvgUtil}(a_m)}$ as $m \to \infty$. To show that distortion is not bounded in $\beta$, it remains to show that $\text{AvgUtil}(a_m) \to 0$. The lemma already tells us that $\text{AvgUtil}(a_t)$ (for $t = 1, 2, \ldots$) is a monotonically decreasing sequence. Since the social welfare is nonnegative, the sequence is bounded from below and thus convergent, which also implies that the sequence of differences $\text{AvgUtil}(a_{t-1}) - \text{AvgUtil}(a_t)$ must converge to 0. Since the bound $\frac{2}{3\beta}\log(1 + \tanh(\beta/4\text{AvgUtil}(a_{t-1}))^3)$ is sandwiched between these differences and 0, it must also converge to 0. Since it is continuous in $\text{AvgUtil}(a_{t-1})$ and positive for all positive values of $\text{AvgUtil}(a_{t-1})$, this implies that $\text{AvgUtil}(a_{t-1})$ must converge to 0. This shows that $\text{AvgUtil}(a_m) \to 0$ for large $m$, and that the distortion grows unboundedly large as $m$ increases, which concludes our proof. □

---

[15]This means that this lower bound fits into the social choice subsetting. Note that the theorem does not apply to Borda count (which is not defined for a general distribution $\nu$), but to RLHF considered as a voting rule.

