# OpenReview forum: "Distortion of AI Alignment: Does Preference Optimization Optimize for Preferences?"
_NeurIPS.cc/2025/Conference — NeurIPS 2025 poster_

### Official Review · Reviewer_qFQ9 · 2025-06-09

**Clarity:** 3
**Significance:** 3
**Originality:** 3
**Rating:** 4
**Confidence:** 3

**Summary:**

This paper investigates the distortion, which is the ratio of the maximum expected utility and the expected utility achieved, of several preference alignment algorithms. The results include the lower bounds and upper bounds of algorithms without/with KL-constraints to a reference policy. Specifically, the paper proves that NLHF achieves the best distortion, while RLHF and DPO suffer a larger distortion.

**Questions:**

Intuitively, what is the reason that NLHF outperforms RLHF? Is it because NLHF optimizes on matrix $M$, which contains more information than the estimated utility vector used by RLHF?

**Ethical Concerns:**

["NO or VERY MINOR ethics concerns only"]

**Final Justification:**

The authors' response address my questions, and I will keep the rating.

**Limitations:**

Yes.

**Quality:**

3

**Strengths And Weaknesses:**

## Strengths

- The writing of the paper is clear and easy to follow.
- The results sound novel to me. This paper shows that the final performance of NLHF (expected utility) might be an exponential multiple of that of RLHF. The reason is that RLHF cannot handle users with heterogeneous feedback since it encodes all preference data into a single utility vector. This further justifies the motivations of work focusing on RLHF with heterogeneous feedback, such as [1] and [2].

## Weaknesses

- This paper does not have any experiments to corroborate the results.
- All results of RLHF are limited to the single-preference RLHF, where the algorithm ignores the source of each preference data. In other words, the RLHF considered in this paper does not consider heterogeneous preferences.

[1] Zhong, Huiying, et al. "Provable multi-party reinforcement learning with diverse human feedback." arXiv preprint arXiv:2403.05006 (2024).

[2] Park, Chanwoo, et al. "Principled rlhf from heterogeneous feedback via personalization and preference aggregation." arXiv e-prints (2024): arXiv-2405.

---

> ### Author Rebuttal · Authors · 2025-07-31
>
> Thank you for your feedback and appreciation of our results. We address your comments below.
>
> > **(Single-preference RLHF)** All results of RLHF are limited to the single-preference RLHF, where the algorithm ignores the source of each preference data. In other words, the RLHF considered in this paper does not consider heterogeneous preferences.
>
> To avoid ambiguity for other readers, let us start by emphasizing that our model does capture preference heterogeneity (in fact, even richer forms of heterogeneity than Park et al. (2024) because we do not place assumptions on users’ utilities), and that the reviewer wishes that we had studied additional alignment methods such as proposed RLHF variants tailored for heterogeneous preferences.
>
> We believe that it is extremely natural for our paper to initiate the study of alignment distortion by investigating RLHF/DPO [Christiano et al., Ziegler et al., Rafailov et al.], the incumbent methods for AI alignment in practice, as well as the minimax optimal alignment method NLHF. Studying the distortion of other alignment methods, including the proposed RLHF variants by Park et al., is an interesting direction for future research. Given that these algorithms are based on assumptions about user preferences (clusterability or a low-dimensional common representation), we would expect the algorithms to obtain very high distortion in our model, in which these assumptions need not hold. These algorithms also require a large number $d$ of comparison data per user to have any chance at clustering or representation learning. By comparison, it is impressive that NLHF can obtain an excellent distortion (optimal under the assumptions given in Thm. 3) already with a single comparison per user.
>
> > **Experiments & Practical implications**
>
> While we agree that empirical evaluations are an intriguing next step of our work, we want to echo reviewer `mjbj` who writes that “asking for extensive empirical results would be outside the scope of the paper (the theoretical results are already compelling as they stand, and the experiments themselves would be expensive enough to deserve their own paper anyways)”. Besides the computational cost of experiments, the limitations of pairwise comparisons we show also mean that most post-training datasets, which consist of such comparisons, are insufficient for such an evaluation. For future work, we are intrigued by recent datasets [e.g., Kirk et al., NeurIPS’24] that include expressions of cardinal information and the rater’s identity, but using such data to evaluate models based on user utilities will require overcoming challenges about users’ inconsistent reporting of cardinal values and the low amount of preference data per user in the datasets.
>
> While our work does not include such experiments, our main contribution is to introduce and analyze distortion as a theoretical metric for worst-case robustness. Our work also carries lessons for the practitioner as described below:
> - **Limitations of reward models as evaluation metrics.** Our analysis shows that a model’s performance according to the reward model is a highly flawed measure of agreement with human preferences, and should not be conflated with average user satisfaction. Our results show that this mismatch does not reduce with increasing the amount or diversity of comparison data, but is a persistent bias due to information-theoretic barriers. Since (ordinal) comparison data is inherently limited in its informativeness about user satisfaction, empirical evaluations of user satisfaction should be based on users’ expression of (cardinal) preference intensity.
> - **Sensitivity to choice of post-training dataset**: Our results demonstrate that the reward-based pipeline of RLHF is sensitive to the sampling distribution over pairwise comparisons, whereas the reward-free method NLHF is more robust and analytically tractable. There is recent interest in using post-training data sets across models and platforms (e.g., Zhang, et al. *“Cultivating Pluralism In Algorithmic Monoculture: The Community Alignment Dataset”*). Our results indicate that practitioners who use RLHF should be wary of lack of knowledge and control over how their data set was curated. As the community moves towards general purpose and public post-training data sets, there is even more argument in favor of using methods that are robust to the distribution of comparison data such as NLHF.
> - **Failure modes that may occur in practice**: As is common for any worst-case analysis, our lower-bound instances are stylized and chosen to be easy to analyze. But from a practical lens, we believe that these instances exaggerate patterns that may well occur in real-world examples (for example, that frequent pairwise comparisons between alternatives suppressed by the reference policy can cause RLHF to misalign on unsuppressed actions). Our theory allows the exploration of such patterns in a systematic way, based on which patterns have the biggest worst-case effect.
>
>
> > **Intuitively, what is the reason that NLHF outperforms RLHF?** Is it because NLHF optimizes on matrix, which contains more information than the estimated utility vector used by RLHF?
>
> We agree with your intuition that RLHF is limited by the fact that its utility vector discards much of the relevant information, and that, conversely, NLHF profits from a richer representation of preferences, which preserves information about disagreements/heterogeneity. In addition, NLHF’s strength comes from its algorithmic approach — its computation of the policy by solving  a 2-player game “hedges” (l. 310) against any other alternative being much better, which makes the distribution particularly robust.

---

> > ### Comment · Reviewer_qFQ9 · 2025-08-02
> >
> > Thank you for your response, and I will keep my rating!

---

### Official Review · Reviewer_i2wL · 2025-06-11

**Clarity:** 2
**Significance:** 2
**Originality:** 3
**Rating:** 4
**Confidence:** 3

**Summary:**

The paper initiates a study of distortion in the alignment setting. Their model assumes a finite alternative set $A$ of size $m$ and, $n$ user each with a utility vector $u_i \in [0,1]^m$ drawn from a common prior distribution $\mathcal{D}$. Pairwise comparison queries are generated by sampling two alternatives i.i.d. from a distribution $\mu$ over $A$. And then all the users provide $d$ noisy comparison according to the Bradley-Terry model with a shared temperature $\beta$.

The authors then distinguish two regimes: one with no KL‐divergence limit on the learned policy, and the other with. They call them social choice v.s. alignment settings.

Both RLHF and NLHF are analyzed in each regime. Their core technical tool is a linearization lemma that bounds the deviation of the win‐rate of $x \succ y$ from a a half by affine functions of $x,y$'s social welafare. The principal conclusion is that NLHF admits strictly tighter upper bounds on distortion than RLHF under both unconstrained and KL‐constrained setups.

**Questions:**

Your contribution is novel and timely. However, I still have several major concerns. I am happy to revise my evaluation once they are addressed:

1. Dependence on m.n.

  In Theorems 2 and 7, if one allows $m \propto n^2$ and $\mu$ to be uniform $\mu(x)=1/m$, the second term in the finite-sample distortion bound appears to grow without bound as $n \to \infty$. Could you discuss how \(m\) and \(n\) must scale relative to one another to ensure the bound remains meaningful? I am a little bit uneasy about the correctness of your statements.

2. Behavior of distortion at extreme $\beta$

  Intuitively, one might expect a U-shaped dependence on $\beta$  : As $\beta \to 0$, comparisons become nearly random, losing information; as $\beta \to \infty$, even a tiny utility gap yields deterministic comparison outcomes (e.g. originally $u_i(x)=0.3; u_i(y)=0.2999$, but a very high $\beta$ would make it almost always the case that $x \succ_i y$), also not reflecting the original preference intensity.
  Yet your distortion bounds only worsen for large $\beta$ , with no analogous degradation at small $\beta$. Could you clarify why the bound does not reflect information loss in the first regime?

3. Reference policy in the KL-ball

   In Theorem 6, instead of exhibiting a single adversarial reference policy for the lower bound, is it possible to bound the distortion between an arbitrary reference policy and all policies within its KL-ball? Also, in Theorem 6 you refer to “a sequence of alignments”; could you clarify the intended meaning and why you chose this phrasing?


4. Role of the post–Theorem 5 discussion

  The paragraph following Theorem 5 appears to critique the distortion framework even this paper is centred about distortion. I didn't fully get why you made this discussion. It was nice that you have already adopted a new, alignment-oriented distortion measure, and drew a line between the traditional setting (as introduced around line 128) to your new alignment setting. Were you just trying to stress that the social choice setting is really not important for the alignment practitioners?

Additionally, I was also a bit disappointed by the paragraph starting from line 128: first, it's very good that you have mentioned the context-dependent aspect of alignment, but then you directly make an assumption instead of discussing the alternatives (even though you have mentioned this as future work in later section). Second, i wished to see more comparisons/motivations for your model v.s. a tranditional social choice model, for example, you make many more distributional assumptions and allow each user to provide only $d=1$ or $2$ comparisons, which differs fundamentally from the full-ranking assumption in traditional social choice.

**Ethical Concerns:**

["NO or VERY MINOR ethics concerns only"]

**Final Justification:**

The reviewers have addressed all of my concerns. I believe some of the writings may still need to be improved, as well as the justification of the model. But it should be accepted.

**Limitations:**

Yes.

**Quality:**

2

**Strengths And Weaknesses:**

Strengths:
1. This paper intiates a series of important questions regarding distortion, and it is outstanding in the sense that it takes social choice literature as inspiration but also goes beyond it--- it takes KL constraints into account.
2. The results are extensive. I particularly like the discussion section as it partially addresses some of the concerns i was having.

Weaknesses:
1. The clarity can be improved:

line 96, maybe stress that : the objective is to find an alternative $x \in A$, because with the original alignment techniques, a policy is able to generate new alternatives. I was unsure which case you are addressing until i read many more lines.

line 115, it took me a while to figure out/guess what you meant by saying the "expectation is taken over the pairwise comparisons".

line 117, 156, the social welfare of a rule not only depends on the rule itself, but also on the pairwise comparisons distribution as well as the users distribution, but both notations really did not reflect this. I understand that the notations would not look so nice, then you could introduce a shorthand.

line 253, maybe change the wording "the distortion guarantee is at least..."

2. Please see my questions.

---

> ### Author Rebuttal · Authors · 2025-07-31
>
> Thank you for your comments and suggestions. We address the questions below.
>
> > **(Dependence on m,n)** In Theorems 2 and 7, if one allows $m\propto n^2$ and $\mu$ to be uniform, the second term in the finite-sample distortion bound appears to grow without bound as $n\to\infty$. Could you discuss how m and n must scale relative to one another to ensure the bound remains meaningful?
>
> Recall that $m$ denotes the number of candidate responses, and $n$ is the number of crowdsourced labelers sampled from the user population. Our analysis focuses on the regime where  $m$ is fixed and finite (e.g., a finite number of potential responses, or bounded latent complexity of the set of responses), while $n$ represents the sample complexity, in other words the total number of labelers sampled from distribution and can be made arbitrarily large by sampling more labelers from the user distribution. As is standard in determining the sample complexity in statistical learning, our bounds are optimized for the regime in which the number of comparisons is large compared to the latent complexity of the space. As a result of this focus, following your assumption that $\mu$ is uniform over $m$ candidates, the finite distortion bound Thm 2 starts having bite when $n\gg m^3\log m$, and the bound in Thm 7 when $n\gg m^2\log m$.
>
> > **(Extreme beta)** Intuitively, one might expect a U-shaped dependence on $\beta$ ... as $\beta\to0$, comparisons become nearly random, losing information …  Could you clarify why the bound does not reflect information loss in the first regime?
>
> Your intuition about the U-shaped curve is correct. For small but nonzero $\beta$, the distortion in the *infinite* sample regime ($n\to\infty$) remains finite and close to $1$, but it is indeed true that the welfare approximation for a given *finite* number of samples would grow. This can be seen in the error terms of Thm 2 and 7: if $\beta \to 0$ as the other quantities remain fixed, the concentration error term would grow and and make the lower bounds on social welfare (for Borda or NLHF) vacuous. This reflects your intuition that each comparison becomes more random and thus contains less information, so achieving finite distortion requires a lot more samples (large $n$) to counteract this randomness. We will add a discussion after the corresponding theorems to clarify this.
>
> > **(KL ball)** In Theorem 6, instead of exhibiting a single adversarial reference policy for the lower bound, is it possible to bound the distortion between an arbitrary reference policy and all policies within its KL-ball?
>
> The question can be potentially interpreted in two ways:
> - (1) Can we avoid the exponential distortion in Thm 6 by requiring $\pi_{ref}$ to have bounded distortion compared to the optimal policy in the KL ball (so that the choice of $\pi_{ref}$ is not fully adversarial)? The answer is no: in our constructed instance, c has welfare on order 1 but can’t be meaningfully used in KL ball; a has welfare $\Theta(1/\beta)$ and b has welfare $\Theta(e^{-\beta})$. Therefore, both $\pi_{ref}$ and $\pi^*$ has $\Theta(1/\beta)$ welfare, which implies that $\pi_{ref}$ already has constant distortion relative to all policies in its KL ball.
> - (2) Can we upper bound the distortion as a function of the KL divergence between $\mu$ and $\pi_{ref}$? This would be an intriguing strengthening of the open question about the $\mu=\pi_{ref}$. However, the same construction shows that such a bound cannot be polynomial: we have $KL(\pi_{ref} \\| \mu)=\Theta(\beta)$ but distortion $=e^{\Omega(\beta)}$.
>
> Despite the above observations, we agree that it is important to understand the distortion of RLHF beyond the specific failure mode constructed in Thm 6. This is an interesting open question for future research.
>
> > In Theorem 6 you refer to “a sequence of alignments”; could you clarify the intended meaning and why you chose this phrasing?
>
> By “sequence of alignment problems” we refer to a sequence of instances with increasing values of beta and m, where the distortion of this sequence goes to infinity. This is standard usage, and we will reword it to make it clearer.
>
> > The paragraph following Theorem 5 appears to critique the distortion framework… It was nice that you have already adopted a new, alignment-oriented distortion measure, and drew a line between the traditional setting (as introduced around line 128) to your new alignment setting. Were you just trying to stress that the social choice setting is really not important for the alignment practitioners?
>
> Thank you for pointing out this ambiguity in the writing. We believe your confusion is due to the fact that we have both:
> a “social choice setting” (l. 112), a special case of our alignment model (l. 128) in which the alignment method can output any distribution over actions, the same output as (randomized) social choice [e.g., 1]
> the setting of “classic deterministic-choice distortion” (l. 162), the setting considered in papers on voting in social choice theory. This setting has the same output as above, but additionally assumes that users deterministically choose the higher-utility action, rather than choosing through a Bradley–Terry model.
> Our discussion starting in l. 254 refers to the latter “classical setting”. Our intent is not to downplay social choice theory’s perspective (or the social choice subsetting), but to highlight that adopting the BT model provides a “beyond-worst-case” lens [2] that is more meaningful not only for alignment applications, but also for the study of voting. For example, a “significant barrier” [3] to the classic deterministic-choice distortion approach is that all known voting rules with low deterministic distortion are highly artificial, whereas the minimax-optimal voting rule for BT distortion is a natural choice in alignment (NLHF) and the voting special case (Maximal Lotteries). We will revise the discussion to resolve this confusion.
>
> [1] Gibbard, Allan. “Manipulation of Schemes That Mix Voting with Chance.” Econometrica 45, no. 3 (1977): 665--681.
> [2] Roughgarden, Tim. Beyond the worst-case analysis of algorithms. Cambridge University Press; 2021 Jan 14.
> [3] Ebadian, Soroush, Anson Kahng, Dominik Peters, and Nisarg Shah. “Optimized Distortion and Proportional Fairness in Voting.” ACM Transactions on Economics and Computation 12, no. 1 (2024): 1–39.
>
> > **the context-dependent aspect of alignment**
>
> Note that all our lower bounds extend to a model that is aware of states, since our setting can be thought of as the special case in which preferences are constant across states. Our NLHF upper bound can also be used to show that this lower bound can be met, in the limit of infinitely many samples, even in a state-dependent model (assuming that preferences are Lipschitz in the state). At a high level, this can be shown with an alignment method that determines the policy at each state $x$ in the support of the state distribution by applying NLHF to only comparison pairs in a small neighborhood of states around $x$. This shows that our state-independent model gives the right infinite-sample behavior for a world with states; some of the authors are currently thinking about how to be more sample efficient in such a setting.
>
> > **more comparisons/motivations for your model v.s. a traditional social choice model**, for example, you make many more distributional assumptions and allow each user to provide only d=1 or 2 comparisons, which differs fundamentally from the full-ranking assumption in traditional social choice.
>
> - **BT vs deterministic comparisons**: We discussed the virtues of modeling choices as Bradley–Terry rather than the classic modeling assumption of deterministic choices, and this is indeed what our discussion after Thm. 5 is meant to convey. We would also like to point out that we do not see deterministic choices as a more realistic model of human behavior than Bradley–Terry, and that our linearization approach should extend to similar random-utility models for pairwise decisions.
>
> - **Pairwise vs full ranking**: Though we focus on the setting in which each voter provides few comparisons for clarity of exposition (a fitting model of current alignment practice), almost all of our results remain intact when voters provide full rankings. Since these rankings are randomized, users’ rankings would be distributed according to a Plackett–Luce model (l. 110), so that each pairwise comparison is still Bradley–Terry distributed. This model fits into the extended model we discuss in l. 354 (by having a constant distribution over $\binom{m}{2}$ pairs), and we discuss in lines 361–363 how the results generalize. In this special case, the Borda rule also continues to be defined, with the same upper and lower bounds (and slightly better finite-sample guarantees for the upper bound).
>
> - **Distributional assumption**: We assume a distribution of user utility functions, and each labeler is an independent sample drawn from this distribution. This formulation allows each user/labeler to have a distinct utility function, rather than restricting the population to a small number of types. This generalizes the traditional social choice setting with is a finite number of voters, as we can take this distribution to be uniform over these voters (there is a minor difference between sampling with or without replacement from the population, which should vanish when the number of samples is large).

---

> > ### Comment · Reviewer_i2wL · 2025-08-02
> >
> > I thank the reviewer for the detailed response. Overall, I think it's a solid paper and I am happy to increase my score.

---

### Official Review · Reviewer_mjbj · 2025-06-30

**Clarity:** 3
**Significance:** 3
**Originality:** 3
**Rating:** 5
**Confidence:** 4

**Summary:**

Consider RLHF as a way of aggregating preferences across a population. Then a basic question which this paper considers is whether current alignment methods such as DPO and RHF reliably lead to high average utility across users. In short, the conclusion is no: stochastic ordinal preferences don't contain enough information to ensure high average utility across a heterogeneous user population.

More specifically, the distortion in the case where there's no KL regularization is at least linear in $\beta$ — that is, the temperature term in the Bradley-Terry model, which quantifies how deterministic user preferences are. When KL regularization is added—so that we're now considering standard ROHF or DPO—the distortion becomes exponential in $\beta$.

In short, the paper presents a strong case that standard alignment methods will fail to respect the preferences of the average user, let alone minority users.

**Questions:**

The confusions I have largely concern what the practical implications of the work are (I found the theorems themselves clear).

1. Suppose you had two users A and B, and use DPO to train a reward model R_mix over a 60/40 mixture of the two user’s preferences. Is it possible to classify R_mix from R_A or R_B (trained on each user’s preferences alone) using the lower bounds on the distortion (Theorems 5, 6)? That is, drawing samples from the original population (mixture for R_mix, A for R_A, etc), computing the sample mean distortion, and classifying based on those distortions. If so, this could potentially serve as a post-hoc measure of how heterogeneous the preferences are.
2. The discussion in Section 5 seems to suggest that simply replacing KL divergence with some other regularization from the reference policy is insufficient to reduce the distortion. But what if multiple reference policies were used (e.g. different base models, or different checkpoints of the same model), and the regularization consisted only of the minimum KL from each reference policy? So long as no reference policy puts a tiny amount of probability mass on c (line 289) wouldn’t this improve the lower bound in Theorem 6?

**Ethical Concerns:**

["NO or VERY MINOR ethics concerns only"]

**Final Justification:**

Authors articulated what the concrete empirical predictions are. Conditional on them adding that to the paper, I am raising my score to 5 tentatively (we'll see how the rest of the discussion period with other authors goes as well).

**Limitations:**

Yes

**Quality:**

4

**Strengths And Weaknesses:**

Strengths:

Overall I find the theoretical argument compelling: the assumptions stay true to the Bradley Terry model and properly incorporate KL regularization, and \emph{lower} bounds on the difference of \emph{average} utility are more compelling than results which only consider the worst case.

Weaknesses:

Yet even after reading the paper carefully, I am left wondering what the actual implication for alignment is? I suspect that this is largely a gap in writing that could be addressed during the discussion period. For instance section 5 seems to suggest that the issue is primarily the presence of KL regularization to a single reference policy. Having a short section which very clearly states for alignment practitioners what the theory suggests should be different would greatly improve the readability of the paper. I think that in this case, asking for extensive empirical results would be outside the scope of the paper (the theoretical results are already compelling as they stand, and the experiments themselves would be expensive enough to deserve their own paper anyways), but \textbf{the paper should at least make the question clear, enabling empirical follow-up research}.

Quality:

The overall quality of the paper is high. The writing is clear, the notation is consistent with other papers on RLHF alignment, and Figure 1 adds a lot of intuitive understanding.

Clarity:

I predict reviewers will be pretty divided on the clarity of the paper, depending on their background. Personally, I think it’s very well written given the subject matter. However, readers from a strong empirical background will likely struggle to identify the thesis, since it’s not presented in a typical "Figure 1" style with bolded performance results that beat a baseline. That’s why I propose including a clear section at the end (and foreshadowed at the beginning) that concludes with a precise prediction for an experiment in a follow-up paper.

Significance:

I think the paper, even in its current form—lacking any crisp predictions—still manages to have moderate significance. It clearly argues that the burden of evidence should be on showing that RLHF or DPO are fair, given that we have every theoretical reason to believe they are not. The significance would be greatly improved by including a single crisp prediction that could be explored empirically, in which case I would raise my score.

Originality:

To the best of my knowledge, the bounds presented here are new.

---

> ### Author Rebuttal · Authors · 2025-07-31
>
> Thank you for your feedback and appreciation of our theoretical results. We address your comments below.
>
> > Practical implications and directions for empirical study
>
> This is an insightful suggestion. We will add a section on the insights our work carries for the practitioners and pointers for future work. The implications we have in mind are the following:
> - **Limitations of reward models as evaluation metrics.** Our analysis shows that a model’s performance according to the reward model is a highly flawed measure of agreement with human preferences, and should not be conflated with average user satisfaction. Our results show that this mismatch does not reduce with increasing the amount or diversity of comparison data, but is a persistent bias due to information-theoretic barriers. Since (ordinal) comparison data is inherently limited in its informativeness about user satisfaction, empirical evaluations of user satisfaction should be based on users’ expression of (cardinal) preference intensity.
> - **Sensitivity to choice of post-training dataset**: Our results demonstrate that the reward-based pipeline of RLHF is sensitive to the sampling distribution over pairwise comparisons, whereas the reward-free method NLHF is more robust and analytically tractable. There is recent interest in using post-training data sets across models and platforms (e.g., Zhang, et al.. *“Cultivating Pluralism In Algorithmic Monoculture: The Community Alignment Dataset”*). Our results indicate that practitioners who use RLHF should be wary of lack of knowledge and control over how their data set was curated. As the community moves towards general purpose and public post-training data sets, there is even more argument in favor of using methods that are robust to the distribution of comparison data such as NLHF.
> - **Failure modes that may occur in practice**: As is common for any worst-case analysis, our lower-bound instances are stylized and chosen to be easy to analyze. But from a practical lens, we believe that these instances exaggerate patterns that may well occur in real-world examples (for example, that frequent pairwise comparisons between alternatives suppressed by the reference policy can cause RLHF to misalign on unsuppressed actions). Our theory allows the exploration of such patterns in a systematic way, based on which patterns have the biggest worst-case effect. These should be fruitful for future empirical research.
>
>
> > (Question 1) Suppose you had two users A and B, and use DPO to train a reward model R_mix over a 60/40 mixture of the two user’s preferences. Is it possible to classify R_mix from R_A or R_B (trained on each user’s preferences alone) using the lower bounds on the distortion (Theorems 5, 6)? That is, drawing samples from the original population (mixture for R_mix, A for R_A, etc), computing the sample mean distortion, and classifying based on those distortions. If so, this could potentially serve as a post-hoc measure of how heterogeneous the preferences are.
>
> We’re not entirely certain if we fully understand the details of this question, but our interpretation is as follows: if we had enough samples from users A, B, and the mixture respectively to train reward models for each, could one compute the distortion of the mixture population and use it as a post-hoc measure of heterogeneity? The answer to this question is positive. However, note that training a reward model for, say, user A, requires a large number of pairwise comparisons, and several independent comparisons for the same pair of actions, to be collected from user A (and the same for user B). This would be possible if we could elicit comparisons from many copies of user A, but deviates from the crowdsourcing setting motivating our work, in which  users need not be partitioned into homogeneous types, and the small number of comparisons provided by each user might not even make it easy to cluster users if they do indeed come from few types. In regimes where it is possible to construct reward models (say, because an algorithm can cluster users into few homogeneous types), distortion could serve not only as a diagnostic for heterogeneity but also as a guide for improving alignment. In particular, optimizing for the mixture of each subgroup’s reward models could potentially yield a model that is better aligned to an average user (than fitting a single reward model over the mixture of preference data).
>
> > (Question 2) what if multiple reference policies were used (e.g. different base models, or different checkpoints of the same model), and the regularization consisted only of the minimum KL from each reference policy? So long as no reference policy puts a tiny amount of probability mass on c (line 289) wouldn’t this improve the lower bound in Theorem 6?
>
> We agree that it is interesting to study whether the pessimistic lower bounds for RLHF can be mitigated by using multiple reference policies, which pushes in a similar line to our open question about $\mu = \pi_{ref}$ in lines 302–304. Note that there are subtle tradeoffs for studying this case: on the one extreme, if all policies place similarly small mass on the critical candidate $c$, then the KL regularization to several reference policies still prevents the output policy from allocating meaningful mass on $c$, and our lower bound construction persists. On the other extreme, if there are many, sufficiently different reference policies, taking the minimum KL may effectively remove the regularization. While this does benefit from our better upper bounds of section 3 in the setting without a KL constraint, it also undermines the purposes of using KL regularization in practice.
>
> Finally, our construction amplifies a phenomenon that might likely appear in practice, if certain types of response (e.g., instruction-following but harmful) are not yet suppressed when the preference data is sampled, but are suppressed by all reference policies. In such a scenario, constraining relative to all reference policies would not circumvent the problem. Conversely, even if all reference policies place high mass on all alternatives with high mass in $\mu$, we do not know if this is enough to ensure a low distortion for RLHF, or if other failure modes exist. Investigating this is an intriguing direction for future research.

---

> > ### Comment · Reviewer_mjbj · 2025-08-01
> >
> > Nice, this (mostly) makes sense and addresses my concerns. Just a few comments:
> >
> > Regarding practical implications and directions for future study, I'll just summarize in my own words to check for understanding:
> > - It sounds like the ideal would be to use NLHF using (cardinal) preference intensity data.
> > - One potential experimental setup: Suppose we do this (on general purpose and public post-training data sets), and seek to compare it with the standard RLHF pipeline on ordinal preferences. Then run a follow-up user study, where users rate the generations produced by models fine-tuned under each pipeline using cardinal preference intensity scores.
> > - Prediction: the average score will be higher on the NLHF pipeline.
> > Did I miss anything? (If not, I encourage you to directly emphasize a prediction like this directly in the section you are adding to the paper for pointers on future work).
> >
> > It sounds like what I proposed in Question 1 isn't a practical post-hoc measure of how heterogeneous the preferences are, because it doesn't really work in the crowdsourced setting.
> >
> > Interesting point on question 2, that if you have a lot of reference policies then it might dilute the effect of KL regularization so much that it has no effect. It does seem like this suggests an interesting follow-up work (outside the scope of this paper) which could explore this further.
> >
> > Please respond to my first paragraph about the potential experimental design, but you have already sufficiently addressed my concerns about practical implications to justify raising my score to 5.

---

> > > ### Author Response · Authors · 2025-08-03
> > >
> > > Thank you for your response and for raising the score. We will make sure to include a discussion of the practical implications in revisions of our paper. Here we address you follow-up questions on the experimental setup:
> > > > It sounds like the ideal would be to use NLHF using (cardinal) preference intensity data.
> > >
> > > NLHF operates on *ordinal* preferences by computing a Nash equilibrium of the marginal preference matrix. Indeed, we show that NLHF is more robust to sampling distributions than RLHF when only ordinal data (i.e., no numerical utilities) is available. However, if all cardinal preferences are present, then the better approach that completely avoids distortion is to directly optimize for the average utility function computed from cardinal data.
> > >
> > > > One potential experimental setup: Suppose we do this (on general purpose and public post-training data sets), and seek to compare it with the standard RLHF pipeline on ordinal preferences. Then run a follow-up user study, where users rate the generations produced by models fine-tuned under each pipeline using cardinal preference intensity scores.
> > >
> > > Yes, we agree that conducting user study is a reasonable way to compare the relative distortion of two alignment methods in practice when we lack access to cardinal utilities. The reviewer’s proposed setup would involve (1) running two alignment methods (NLHF vs RLHF) on general-purpose ordinal datasets, and (2)  running a follow-up user study in which users provide cardinal ratings of the outputs from both methods. This allows us to estimate the social welfare of the NLHF or RLHF policies, thereby comparing their distortions on the same instance. A small caveat is that this approach only measures the *relative distortion* (i.e., average user satisfaction for NLHF vs RLHF), however, this seems like a worthwhile goal in its own right. Computing absolute distortion of the two methods would potentially require more cardinal data on all possible alternatives to identify the optimal policy and its welfare, which can potentially be costly.
> > >
> > > > Prediction: the average score will be higher on the NLHF pipeline. Did I miss anything? (If not, I encourage you to directly emphasize a prediction like this directly in the section you are adding to the paper for pointers on future work).
> > >
> > > This aligns with our intuition as well based on our results for specifically constructed instances. We’re unable to predict whether NLHF outperforms RLHF on every instance, but when the dataset exhibits patterns such as frequent pairwise comparisons between alternatives suppressed by the reference policy, our intuition is that NLHF would achieve a higher user average score than RLHF.

---

> > > > ### Comment · Reviewer_mjbj · 2025-08-05
> > > >
> > > > Thank you for these clarifications.

---

### Official Review · Reviewer_DYaW · 2025-07-03

**Clarity:** 3
**Significance:** 3
**Originality:** 3
**Rating:** 4
**Confidence:** 3

**Summary:**

This paper introduces "distortion" as a novel framework to theoretically analyze the performance of AI alignment methods when faced with diverse and conflicting human preferences. Distortion quantifies the worst-case gap between the optimal average utility achievable and the utility attained by a given alignment method, which learns from probabilistic pairwise comparisons modeled by the Bradley-Terry model. The research provides a comparative analysis of major alignment techniques, demonstrating that NLHF is robust and achieves the minimax optimal distortion guarantee of $(1/2+o(1))\beta$. In contrast, methods like RLHF and DPO are shown to suffer from significantly worse distortion, which can become exponential or even unbounded in the general alignment setting that includes a KL-divergence constraint from a reference policy. Ultimately, the work establishes that a certain level of distortion is unavoidable, revealing a fundamental information bottleneck when relying solely on ordinal feedback from a heterogeneous population.

**Questions:**

See Weakness.

**Ethical Concerns:**

["NO or VERY MINOR ethics concerns only"]

**Final Justification:**

I maintain my original score as weakly accept.

**Limitations:**

Yes.

**Paper Formatting Concerns:**

No.

**Quality:**

4

**Strengths And Weaknesses:**

### Strengths
The paper's findings are significant because they draw a sharp theoretical distinction between different alignment approaches. The result that RLHF/DPO can have exponential or even unbounded distortion under certain conditions, while NLHF remains optimally robust, is a powerful and potentially impactful conclusion. This provides strong theoretical justification for exploring alternatives to the current RLHF paradigm.



### Weakness
- RLHF results rely on a significant mismatch between the distribution of comparison pairs $\mu$ and the reference policy $\pi_{\rm ref}$, the authors should justify this assumption on real-world problems.
- The paper is entirely theoretical. While this is not inherently a weakness for a theory paper, the lack of any experiments means the practical implications of the distortion bounds are left to speculation. Empirical studies could help determine if the identified failure modes for RLHF occur in practice or if the constants in the distortion bounds (e.g., $\beta$) lead to meaningful differences on real-world problems.

---

> ### Author Rebuttal · Authors · 2025-07-31
>
> Thank you for your feedback and appreciation of our theoretical results. We address your comments below.
>
>
> > (**RLHF lower bounds**) RLHF results rely on a significant mismatch between the distribution of comparison pairs and the reference policy; the authors should justify this assumption on real-world problems.
>
> We have 3 lower bounds on RLHF and not all rely on the mismatch between $\pi_{ref}$ and $\mu$. Below, we provide the intuition behind each construction and their practical relevance.
>
> - (1) Thm. 5 presents an instance where RLHF distortion is lower bounded by $(1-o(1))\beta$  even in the unconstrained setting where proximity constraint to $\pi_{ref}$ is not binding. For example, $\pi_{ref}$ and $\mu$ could both be chosen to be uniform in this case. This establishes a constant factor separation between the distortion of RLHF and NLHF.
>
> - (2) Thm. 6 shows that under KL constraints, RLHF distortion can grow as $e^{\Omega(\beta)}$. The constructed instance does indeed rely on a large mismatch between $\pi_{ref}$ and $\mu$. Of course, a worst-case instance exaggerates problematic properties of the instance to a severity that is unlikely to be encountered by chance, but we believe that the less severe versions of the same pattern may well arise in practice.
>
> Our constructed instance exploits that the comparison distribution $\mu$ often produces a particular type of response, which has very low probability mass in the reference policy. In practice, this type may for example correspond to some form of instruction-following but harmful responses, which is not yet suppressed when the comparison data is sampled, but is suppressed by the reference policy ($\pi_{ref}$) due to guardrails and supervised fine-tuning. More broadly, a mismatch between $\mu$ and $\pi_{ref}$ arises whenever alignment relies on off-policy preference data: for example, Stiennon et al. (NeurIPS’20, “Learning to summarize with human feedback”) sampled candidate summaries for comparison “from several sources including the current policy, initial policy, original reference summaries and various baselines”; other models are trained using pre-existing comparison data, such as the OpenAssistant [Köpf et al. NeurIPS’23] dataset in which candidate responses are written by crowdworkers, which also implies that the data distribution and reference policy may be quite different.
>
> - (3) Thm. 9 gives an unbounded distortion with correlated sampling models, where there is no single distribution $\mu$, and the challenge is about correlations rather than different levels of overall representation of alternatives in $\mu$ and $\pi_{ref}$. Specifically, the instance constructs a sequence of actions and assumes that adjacent action pairs are more frequently compared than non-adjacent pairs. Milder forms of these correlations could for example occur if actions are sampled over time during the policy’s training (where the policy's distribution shifts from actions early in the sequence to actions late in the sequence over the course of training). This result highlights RLHF’ lack of robustness to non-i.i.d. sampling distributions, which again may be present in externally procured comparison datasets.
>
> > (**Experiments & Practical implications**) The paper is entirely theoretical… Empirical studies could help determine if the identified failure modes for RLHF occur in practice or if the constants in the distortion bounds (e.g., beta) lead to meaningful differences on real-world problems.
>
> While we agree that empirical evaluations are an intriguing next step of our work, we want to echo reviewer `mjbj` who writes that “asking for extensive empirical results would be outside the scope of the paper (the theoretical results are already compelling as they stand, and the experiments themselves would be expensive enough to deserve their own paper anyways)”. Besides the computational cost of experiments, the limitations of pairwise comparisons we show also mean that most post-training dataset, which typically consist of such comparisons, are insufficient for such an evaluation. For future work, we are intrigued by recent datasets [e.g., Kirk et al., NeurIPS’24] that include expressions of cardinal information and the rater’s identity. Using such data to evaluate models based on user utilities will require overcoming challenges about users’ inconsistent reporting of cardinal values and the low amount of preference data per user in the datasets.
>
> While our work does not include such experiments, our main contribution is to introduce and analyze distortion as a theoretical metric for worst-case robustness. Our work also carries lessons for the practitioner as described below, which can motivate insightful empirical studies:
> - **Limitations of reward models as evaluation metrics.** Our analysis shows that a model’s performance according to the reward model is a highly flawed measure of agreement with human preferences, and should not be conflated with average user satisfaction. Our results show that this mismatch does not reduce with increasing the amount or diversity of comparison data, but is a persistent bias due to information-theoretic barriers. Since (ordinal) comparison data is inherently limited in its informativeness about user satisfaction, empirical evaluations of user satisfaction should be based on users’ expression of (cardinal) preference intensity.
> - **Sensitivity to choice of post-training dataset**: Our results demonstrate that the reward-based pipeline of RLHF is sensitive to the sampling distribution over pairwise comparisons, whereas the reward-free method NLHF is more robust and analytically tractable. There is recent interest in using post-training data sets across models and platforms (e.g., Zhang, et al.. *“Cultivating Pluralism In Algorithmic Monoculture: The Community Alignment Dataset”*). Our results indicate that practitioners who use RLHF should be wary of lack of knowledge and control over how their data set was curated. As the community moves towards general purpose and public post-training data sets, there is even more argument in favor of using methods that are robust to the distribution of comparison data such as NLHF.
> - **Failure modes that may occur in practice**: As is common for any worst-case analysis, our lower-bound instances are stylized and chosen to be easy to analyze. But from a practical lens, we believe that these instances exaggerate patterns that may well occur in real-world examples (for example, that frequent pairwise comparisons between alternatives suppressed by the reference policy can cause RLHF to misalign on unsuppressed actions). Our theory allows the exploration of such patterns in a systematic way, based on which patterns have the biggest worst-case effect.

---

> > ### Comment · Reviewer_DYaW · 2025-08-07
> >
> > Thanks for your reply. I will maintain my score.

---

### Decision · Program_Chairs · 2025-09-17

**Decision:**

Accept (poster)

**Comment:**

This paper introduces distortion as a rigorous theoretical framework to evaluate alignment methods under heterogeneous human preferences, drawing from social choice theory. The authors provide tight upper and lower bounds showing that standard approaches such as RLHF and DPO can incur large, even unbounded distortion, while Nash Learning from Human Feedback achieves minimax-optimal robustness. Reviewers recognized the novelty and importance of this contribution, praising its strong theoretical results, clarity of analysis, and implications for pluralistic alignment. Concerns centered on the lack of empirical validation, the clarity of presentation in some sections, and the need for more explicit guidance for practitioners. The authors addressed these points effectively in rebuttal, clarifying technical details, explaining the relevance of their lower-bound constructions, and adding a section on practical implications and empirical directions. Several reviewers raised their scores after discussion, acknowledging that while the work is purely theoretical, its contributions are substantial and timely for the alignment community.